
# Assessment of geomorphic effectiveness of controlled floods in a braided river using a reduced-complexity numerical model

Luca Ziliani[1], Nicola Surian[1*], Gianluca Botter[2], Luca Mao[3]

[1] Department of Geosciences, University of Padova, Italy
[2] Department of Civil, Environmental and Architectural Engineering, University of Padova, Italy
[3] School of Geography, University of Lincoln, United Kingdom

*Correspondence to*: Nicola Surian (nicola.surian@unipd.it)

**Abstract.** Most Alpine rivers have undergone strong alteration of flow and sediment regimes. These alterations have notable effects on river morphology and ecology. One option to mitigate such effects is the flow regime management, specifically by the re-introduction of channel-forming discharges. The aim of this work is to assess the morphological changes induced in the Piave River (Italy) due to two different controlled flood strategies, the first characterized by a single artificial flood per year and the second by higher magnitude, but less frequent, floods. The work was carried out applying a 2D reduced-complexity morphodynamic model (CAESAR-LISFLOOD) to a 7 km-long reach, characterized by a braided pattern and highly regulated discharges. The numerical modelling allowed the assessment of morphological changes for four long-term scenarios (2009-2034). The scenarios were defined taking into account the current flow regime and the natural regime, which was estimated by a stochastic physically-based hydrologic model. Changes in channel morphology were assessed by measuring active channel width and braiding intensity. Comparing controlled flood scenarios to a baseline scenario (i.e., no controlled floods) it turned out that artificial floods had small effects on channel morphology. The highest channel widening (13.5%) was produced by the release strategy with higher magnitude floods, while the other strategies produced lower widening (8.6%). Negligible change was observed in terms of braiding intensity. Results pointed out that controlled floods may not represent an effective solution for morphological recovery in braided rivers strongly impacted in their flow and sediment regimes.

## 1 Introduction

Human activities in riverine areas (i.e., river damming, river engineering, gravel mining, land-use change in the drainage basin) have historically led to notable changes in the flow regime (Gore and Petts, 1989; Poff et al., 1997; Magilligan and Nislow, 2005; Poff et al., 2007; Zolezzi et al., 2011; Magilligan et al., 2013; Ferrazzi and Botter, 2019), along with ecological (Collier, 2002; Céréghino et al., 2004; Paetzold et al., 2008; McDonald et al., 2010; Overeem et al., 2013; Espa et al., 2015) and geomorphic functioning of river systems (Hicks et al., 2003; Petts and Gurnell, 2005; Melis, 2011; Ziliani and Surian, 2012; Magilligan et al., 2013; Mueller et al., 2014; Lobera et al., 2016). Nowadays, dam construction is considered a viable strategy to support the increasing energy and water demands due to climate change and population growth (World



Bank, 2009; Lehner et al., 2011). As outlined by Overeem et al. (2013), large reservoirs with a volume greater than 0.5 km$^3$ intercept globally more than 40% of river discharge and ~26% of the sediments transported by the rivers, reducing the global sediment delivery to oceans, and commonly leading to coastal erosion.

Several metrics have been developed to assess both magnitude and temporal trend of alterations of flow regime in rivers

induced by hydraulic infrastructures (Richter et al., 1996; Richter et al., 1997; Martínez Santa-María et al., 2008; Yin et al., 2015), and extensive sediment monitoring efforts or sediment budget estimations have quantified sediment flux alterations (Surian and Cisotto, 2007; Schmidt and Wilcock, 2008; Melis, 2011; Trinity Management Council, 2014; Espa et al., 2015). Several studies documented hydrologic impacts following the extensive realization of dam systems, in particular in the Alpine region over the 20$^{th}$ century (Botter et al., 2010; Comiti, 2012; Bocchiola and Rosso, 2014). Overall, flow regime

alteration has implied significant changes in flow magnitude, frequency, timing and duration, and thermo-peaking phenomena (Gore and Petts, 1989; Frutiger, 2004; Zolezzi et al., 2009; Zolezzi et al., 2011). Impacts on sediment fluxes have been assessed on reaches directly impacted by damming (Graf, 1980; Williams and Wolman, 1984; Gaeuman et al., 2005; Petts and Gurnell, 2005; Schmidt and Wilcock, 2008; Grant, 2012) as well as in lowland gravel-bed rivers affected by cascading connected reservoirs and other human disturbances at the basin scale (Rinaldi and Simon, 1998; Surian and

Rinaldi, 2003; Bilotta and Brazier, 2008; Surian, et al., 2009; Zawiejska and Wyżga, 2010; Ziliani and Surian, 2012; Scorpio et al., 2015).

Overall, it is widely acknowledged that a reduced sediment flux due to dam construction or sediment supply alteration at the basin scale (e.g. due to afforestation, torrent-control works) produces channel changes (namely, narrowing, incision, braiding intensity reduction), and coarsening of bed sediment. Since the 1970s, growing attention was paid to environmental effects

of large dams (Turner, 1971; Vörösmarty et al., 2003). Different river management strategies have been adopted to address dam-related impacts using structural or operational strategies (e.g., Kondolf et al., 2014), or process-based approaches oriented toward restoring water and sediment fluxes (Wohl et al., 2015a). Flow releases from dams have been eventually regulated to reproduce some aspect of the natural regimes (flow and sediment), via seasonal baseflow increase or recovery (McKinney et al., 2001; Sabaton et al., 2008), control in timing and recession rates of releases (Rood et al., 2003; Shafroth et

al., 2010), artificial gravel augmentation or sediment bypassing (McManamay et al., 2013; Kondolf et al., 2014), flood releases (Collier, 2002; Dyer and Thoms, 2006) or high-flow experimental releases (Melis, 2011; Olden et al., 2014). In other cases, the management strategies focused directly on morphological features recovery through dam removal (Poulos et al., 2014; O'Connor et al., 2015), or mechanical vegetation removal (Environment Canterbury Regional Council, 2015).

Environmental flow management plans aim to mitigate some undesired channel adjustments due to dam operations. Due to

the cost of these programs, decision-makers are increasingly requesting the scientific community to develop appropriate tools able to (i) identify and control the factors that cause channel alterations and (ii) to assess effectiveness of management programs. Environmental agencies in several countries require dam operations to respect releasing protocol in an attempt to mitigate adverse impacts on downstream ecosystems (Schmidt and Wilcock, 2008; Olden and Naiman, 2010; Watts et al., 2011; Konrad et al., 2012). Beisdes some successful empirical experiences (Souchon et al., 2008; Konrad et al., 2011),



robust predictive tools and models are becoming much more impellent to predict channel response to dam operations and
      interruption of the longitudinal river continuum (Bliesner et al., 2009; McDonald et al., 2010; Melis, 2011; Coulthard and
      Van De Wiel, 2013; Gaeuman, 2014).

      The assessment of future evolutionary trajectory of channel morphology may be achieved using conceptual models (e.g.,
      Channel Evolution Models – CEMs, as described in Schumm et al., 1984; Simon and Hupp, 1986; Simon, 1989), empirical
(Lane, 1955; Schumm, 1977; Rhoads, 1992) or numerical models, either Computational Fluid Dynamic (CFD) models or
      Reduced Complexity Models (RCM). Previous applications of RCMs on braided rivers have focused mainly on theoretical
      scale-independent analysis (Murray and Paola, 1994), laboratory experiments (Doeschl-Wilson and Ashmore, 2005; Doeschl
      et al., 2006; Nicholas, 2010), or short gravel-bed river reaches (Coulthard et al., 2002; Thomas and Nicholas, 2002;
      Coulthard et al., 2007; Thomas et al., 2007; Van De Wiel et al., 2007). In this study, such as in Ziliani et al. (2013) and
Ziliani and Surian (2016), an attempt has been made to apply a RCM model at mesospatial (i.e., 5-50 km) and mesotemporal
      (i.e., 10–100 years) scales. In particular, the CAESAR – LISFLOOD model (Bates et al., 2010; Coulthard et al., 2013) has
      been herein applied to a 7 km long braided reach of the Piave River (Italy), one of the most heavily and historically regulated
      river system in Italy.

      We applied CAESAR – LISFLOOD model (hereafter C-L) to assess the morphological effects related to two different kinds
of flow regime management strategies: the first characterized by yearly controlled floods with peaks able to transport
      sediments; the second with more infrequent and higher magnitude floods (i.e., floods with recurrence interval equal to 5
      years) released only when notable channel narrowing is observed in the evolutionary trajectory. Both strategies have been
      developed according to two main criteria: (i) maximize the flow regime "re-naturalization", meaning that the "Controlled
      Flood" (CF) duration has to be set in order to increase its yearly likelihood to occur approaching the natural scenario
condition as much as possible; (ii) the "feasibility" of the strategy, verified by the fact that the cumulative volume released
      per year has to be lower than the maximum volume stocked in the reservoirs existing upstream of the study reach.

      This paper aims to address two main issues: (i) to what extent controlled floods can be effective for the geomorphic recovery
      of a strongly regulated braided river? (ii) can the reduced-complexity morphodynamic model CAESAR-LISFLOOD be
      considered a suitable and reliable tool to reproduce the morphological evolution of a large gravel bed river at the given
mesoscales? In the first section of the paper we provide a brief description of the studied river reach. The second section
      presents the available data, the two models used (i.e., the morphodynamic model CAESAR-LISFLOOD, (Bates et al., 2010;
      Coulthard et al., 2013), and the hydrological model, (Botter et al., 2007) and the criteria adopted for the scenario strategy
      design. The third section presents the results concerning (i) the historical river reach morphological adjustments, (ii) the flow
      regime alteration, (iii) the CAESAR-LISFLOOD calibration and (iv) the simulations of three different "Controlled Floods"
(CFs) releases scenarios. Finally, we critically discuss results and examine strengths and weaknesses of CAESAR-
      LISFLOOD and effectiveness of the flow management strategies under investigation.





## 2 General setting of the study area

### 2.1 The Piave River basin

The Piave River is located in north-eastern of Italy, and it flows for about 220 km from the Alps to the Adriatic Sea (Fig. 1).
The basin area is about 3,900 km$^2$, and its average elevation is about 1,300 m a.s.l. (maximum elevation is 3,364 m a.s.l.).
The climate is temperate-humid with an average annual precipitation of about 1,350 mm. Significant annual variations in the
rainfall amount have been measured over the 20$^{th}$ century, but without any statistically relevant trends (Surian, 1999).

As most of the Alpine Italian rivers (Surian and Rinaldi, 2003; Surian et al., 2009; Comiti, 2012), the Piave River has
suffered heavy human impact, which altered the basin and the river channel dynamics (Surian, 1999; Botter et al., 2010;
Comiti et al., 2011; Comiti, 2012). Especially during the 20$^{th}$ century, the Piave basin has experienced a rapid increase of
anthropogenic exploitation by the construction of a series of dams and reservoirs (nowadays there are 13 major reservoirs,
(Botter et al., 2010) built along the main stem and some tributaries from the 1930s to 1960s. At present, a complex
regulation scheme exists (for details see Surian, 1999; Botter et al., 2010), designed to maximize production of hydroelectric
power and the provision of irrigation water (Fig. 1). Flow regulation alters both the flow duration characteristics and volume
of annual runoff in the river. The reservoirs and diversions along the river and its tributaries also affect sediment transport
and supply.

The Piave basin had also experienced historically strong changes due to land use modifications. Especially after the 1950s,
the development of industry and tourism boosted the abandonment of traditional agricultural and cropping activities on the
mountain slopes, causing natural reforestation in the upper parts of the basin (Del Favero and Lasen, 1993). In addition to the
reductions in sediment supply due to trapping by dams and reforestation, intense in-channel gravel mining has also
contributed to alter sediment fluxes since the 1960s. Furthermore, human pressure on the river channel dynamics resulted
from construction of bank protection structures and torrent control works. As a result of these bank protection works, at
present the river can still move laterally, although the available width for planform shifting is narrower than its natural
braided belt.

### 2.2. Study reach

The study reach is ~7 km long (Fig. 1) and is located between Ponte nelle Alpi and Belluno (the drainage area at Belluno is
1,826 km$^2$). In this reach the morphology is mainly braided and wandering. The average slope of the reach is 0.47%, and the
median surface grain size ranges between 18 and 32 mm (Tomasi, 2009). The active channel width ranges between 43 and
452 m, being 241 m on average, while the fluvial corridor width, defined by the presence of Holocene fluvial terraces, ranges
between 106 and 1,110 m, being 672 m on average. Previous studies (Surian, 1999; Surian, et al., 2009; Comiti et al., 2011;
Picco et al., 2016) have outlined that, over the last 200 years, the study reach have undergone notable lateral adjustments
(narrowing up to 66%), but not significant changes of channel pattern. In terms of bed-level changes, two phases have been





identified: a first phase of moderate incision (1970-1990s) followed by a more recent phase (1990s-2003/2007) during which the river has exhibited equilibrium or slight aggradation (Surian, et al., 2009; Comiti et al., 2011).

**3. Materials and methods**

**3.1 Channel morphology and reconstruction of its evolutionary trajectory**

Channel morphology was analysed in order to gather (i) input data for CAESAR – LISFLOOD model, (ii) data for model calibration and (iii) evidence of the evolutionary trajectory of the study reach. River channel, islands, flowing channel, bank protection structures and groynes were digitized using the available aerial photos and terrain models covering the study reach

(i.e., 2003, 2009 - Table 1). The analysis was carried out using ArcGIS 10.2. The flowing channels and the unvegetated or sparsely vegetated bars were merged to obtain measurement of channel width. Braiding index was calculated using the average number of anabranches across the river (Ashmore, 1991; Egozi and Ashmore, 2008). The historical analysis carried out by Comiti et al. (2011), which covered the period 1805 - 2006 has been extended up to 2009.

A LiDAR Digital Elevation Model (DEM) was provided by the Autorità di Bacino delle Alpi Orientali. It was created using

an airborne LiDAR survey that was acquired in July 2003 (orthometric elevations adopted, vertical error estimate ±20 cm) almost contemporary to one of the aerial photos used in this study (Table 1). Even though the river reach is characterized at low flow by the presence of rather small inundated areas, it was not possible to obtain bed elevation in the flowing channel areas with the standard LiDAR data. Therefore, to complete the bed elevation extraction, the water depth was estimated through the application of the method proposed by Bertoldi et al. (2011) using the 2003 aerial photos. This is an optical

remote sensing technique (Marcus, 2012) for retrieving shallow water depth information using the color of the pixel, as Legleiter et al. (2009) demonstrated that the log transformation of the green over red band ratio correlates linearly with water depth across a wide range of substrate types. The linear regression usually should be calibrated by direct measurements of water depths at the time of the aerial survey. Since such data were not available, we calibrated the regression coefficients by referring to the topography of both 2003 and 2009 cross section surveys.

Sediment grain sizes were surveyed in 2009 (Tomasi, 2009) using volumetric sampling of the surface layer (Fig. 1). A single probability density curve was extracted ($D_{50} \sim 24.5$ mm, $D_{15} \sim 2$ mm, $D_{84} \sim 77$ mm) with nine size classes, as required by the C-L morphodynamic model.

**3.2 Analysis of hydrologic regime in the Piave River**

A variety of approaches is available to analyse the impact of river regulation on the natural flow regime of rivers (Richter et

al., 1996; Richter et al., 1997; Martínez Santa-María et al., 2008; Yin et al., 2015). In the case of the Piave River, although several studies have investigated the degree of its hydrological regime alteration (Villi and Bacchi, 2001; Botter et al., 2010; Comiti et al., 2011), such analysis was hampered by (i) the unavailability of a long term flow data series and (ii) the difficulty in sorting between natural and artificial component of the flow regime. In Da Canal et al. (2007) and Comiti et al.





(2011), flow records derived from two gauging stations (Busche and Segusino; Fig. 1), were modified using a specific
corrective factor (Villi and Bacchi, 2001) and then merged. Comiti et al. (2011) confirmed that the largest flood event at
Busche (Fig. 1) occurred in 1966 and reached almost 4,000 $m^3s^{-1}$. Furthermore, their analysis showed that the discharge with
Recurrence Interval (RI) of 2 years was not statistically different if calculated separately for pre- and post-regulation periods
(1954 was used as separation date between the periods). However, higher frequency events (RI ≤ 1.5 year) show a reduction
of peak discharge after 1954. Similar outcomes have also been reported in Picco et al. (2016).

A more detailed analysis of the impact of regulation on river regimes has been presented by Botter et al. (2010), who applied
a physically-based modelling approach to assess the alterations of the streamflow regime observed in various cross sections
of the drainage network downstream of dams and weirs. The authors have applied an analytical stochastic model (Botter et
al., 2007) to characterize the streamflow probability density function (pdf) by means of climate, soil and vegetation
parameters. After a preliminary model application to smaller, unregulated sub-catchments (that allowed to properly verify
the capability of the model to reproduce locally the natural streamflow regime) the authors have applied the model also in
several regulated sections of the Piave River, including Soverzene (about 5 km upstream of Ponte nelle Alpi, Fig. 1), in order
to evaluate the natural flow regime in regulated cross sections and, by difference, the effect of regulation on the statistical
features of the hydrograph. The approach conceptualizes the dynamics of daily streamflow as a sequence of peaks in
response to rainfall and decays in between these jumps. These jump-decay dynamics are then linked to a catchment-scale
soil–water balance where the input is represented by stochastic daily rainfall. In this setting, flow-producing rainfall events
(that lead to streamflow jumps) result from the censoring operated by catchment soils on daily rainfall, and they are modeled
as a marked Poisson process with mean depth α and mean frequency λ. The parameter α identifies the average intensity of
daily rainfall events, while λ is the frequency of flow-producing events, which is smaller than the underlying precipitation
frequency because of the effect of soil moisture dynamics and evapotranspiration. As a consequence, several climate
variables (such as rainfall attributes), as well as soil and vegetation properties were embedded in λ. Additionally, in that
framework streamflow recessions in between flow pulses are assumed as exponential with a mean rate equal to k, which
defines the inverse of the time scale of the hydrological response (i.e., the mean water retention time in the upstream
catchment). Under these assumptions it can be shown that the steady-state pdf of the specific daily discharge (discharge per
unit catchment area) is a Gamma distribution with shape parameter λ/k and scale parameter αk. The model is applied at the
seasonal timescale, and then the annual pdf is calculated as the average of the four seasonal distributions. During winter, the
presence of snow dynamics in the uppermost regions of the catchment is accounted for by reducing the size of the active
contributing catchment and increasing the recession rates as described by Schaefli et al. (2013), with an elevation threshold
of about 1,900 m a.s.l. In spring, a base flow value is added to the modeled streamflow distribution, which corresponds to a
rigid rightward shift of the pdf. The probability distribution of the natural daily streamflows estimated by the model is then
compared to the pdf of the observed daily flows to assess the extent of the impact of regulation in the lower reaches of the
Piave River, and to get some guidelines for devising meaningful strategies of the flow regime management. In particular, the
daily streamflow series used in this study has been recorded from 1995 to 2009 at Belluno gauging station located at the





downstream section of the study reach (Fig. 1). The highest flood event peaks observed in the reference periods (1996, 2000 and 2002) were checked and modified combining data at Belluno and discharge measurements at Soverzene weir (Braidot, 2003).

### 3.3 CAESAR-LISFLOOD model

Over the last 20 years the application of hydro-morphodynamic physically-based numerical models (generally known as Computational Fluid Dynamic models, CFD) mainly focused on laboratory idealized channel configurations (Wu et al., 2000; Defina, 2003; Rüther and Olsen, 2005; Abad et al., 2008) or referred to the morphological dynamic of natural channels over short time periods (Darby et al., 2002; Chen and Duan, 2008; Li et al., 2008; Wang et al., 2008; Zhou et al., 2009). Although their recent development, the restriction of their field of application reflects unsolved issues in terms of data availability and high computational demands (Escauriaza et al., 2017). Only a few recent works (i.e., Nicholas, 2013a; Williams et al., 2016) have shown that CFD models can be applied at larger spatial and temporal contexts. This limitation has driven to develop two-dimensional alternative models that have been commonly referred to as cellular automata (Murray, 2007), cellular models (Murray and Paola, 1994; Coulthard et al., 2002; Thomas and Nicholas, 2002; Coulthard et al., 2007; Parsons and Fonstad, 2007; Van De Wiel et al., 2007), exploratory models, and reduced-complexity models (RCM – Murray, 2007; Nicholas et al., 2006; 2012). These models have a common solution to the problem that is the adoption of simplified hydrodynamic and sediment transport equations derived by the abstractions of the governing physics.

A major advantage of RCMs is their computational efficiency that allows to simulate river evolution over historic and Holocene timescales (e.g., Coulthard et al., 2002; Coulthard et al., 2005; Nicholas and Quine, 2007; Thomas et al., 2007; Van De Wiel et al., 2007). However, the physical realism of such models has received relatively little attention (Nicholas, 2009; Nicholas, 2013b; Ziliani et al., 2013), and only few studies have shown that these models are highly sensitive to the grid resolution of the computational domain (Doeschl-Wilson and Ashmore, 2005; Doeschl et al., 2006; Nicholas and Quine, 2007). Despite the progress shown in several works (Nicholas, 2009; Nicholas, 2013a), there are still few applications in natural rivers characterized by complex channel morphology (Ziliani et al., 2013; Ziliani and Surian, 2016).

In this study, CAESAR–LISFLOOD model (Coulthard et al., 2013) has been applied (1.2 version, reach mode; see "Supplementary material" file for details about the model)). This model is an integration of LISFLOOD-FP (Bates et al., 2010) and CAESAR (Coulthard et al., 2007; Van De Wiel et al., 2007) models. The C-L model links the hydraulics of the former with the erosion and deposition components of the latter. LISFLOOD-FP is a one-dimensional inertial model derived from the full shallow water equations that is applied in the x and y directions to simulate two-dimensional flow over a rasterized spatial domain (Bates et al., 2010). LISFLOOD-FP has been successfully tested to simulate hydraulics in shallow water environments affected by a strongly unidirectional flow (Bates et al., 2010; Neal et al., 2011; Coulthard and Van De Wiel, 2013; Lewis et al., 2013; Skinner et al., 2015; Wong et al., 2015) and for flood inundation simulations characterized by rapid wetting and drying condition (Bates et al., 2010). CAESAR model (Coulthard et al., 2007; Van De Wiel et al., 2007; Ziliani et al., 2013) represents the morphodynamic component of the C-L integrated model. Ziliani et al. (2013) submitted



CAESAR to a rigorous and objective performance evaluation procedure, and showed that (i) CAESAR can be a very powerful tool for modelling spatial and temporal scales still hardly supported by 2D – 3D CFD morphodynamic models, (ii) it can be very useful for setting "what-if scenario" strategies over meso spatial and temporal scales and (iii) it provides reliable bedload sediment budget estimations. From a morphological point of view, Ziliani et al. (2013) have shown that

CAESAR is able to reproduce the average change in channel width, but it performed poorly in reproducing the braided in-channel pattern dynamic and the typical topographic complexity of a braided river at low water stages (e.g., braiding intensity). LISFLOOD-FP and CAESAR have been efficiently integrated and tested into the new CAESAR–LISFLOOD (see Coulthard et al., 2013 for details). Herein the hydraulic element embedded into the model has been verified to be consistent with the LISFLOOD-FP developed by Bates (2010), but the geomorphic component of the C-L model has not

been fully evaluated referring to real case study data. The embedded erosion and deposition modules have been assessed only through the intercomparison of CAESAR and C-L sediment yield results.

### 3.4 Morphodynamic model performance assessment

Several works (Darby and Van De Wiel, 2003; Hoey et al., 2003; Wilcock and Iverson, 2003) emphasized the challenge of a proper calibration of process-based models in fluvial geomorphology due to the increase of uncertainty proportionally with

the complexity of the modeled processes and the number of parameters to be estimated (Formann et al., 2007; Papanicolaou et al., 2008). Despite this intrinsic complexity, it is crucial to understand limitations and performance of RCMs (Aronica et al., 2002; Hall et al., 2005; Lane, 2006), adopting methods that (i) are able to include all the limitations inherent in calibration of this type of model and (ii) are mainly based on field and remote sensing data (Nicholas, 2010). There are currently no international standard methods for the calibration and validation of fluvial morphodynamic models (Mosselman,

2012), and those previously proposed are typically designed for hydrodynamics CFD models (ASME, 1993; Lane et al., 2005). Furthermore, the calibration of RCMs has to be performed keeping in mind that a calibrated model can just be "empirically adequate" (Van Fraassen, 1980) and its validation is just a "confirmation" (Oreskes et al., 1994) that cannot be considered conclusive (Haff, 1996; Lane et al., 2005; Murray, 2007).

In light of all the issues above, the C-L was calibrated referring to the July $5^{th}$ 2003 – August $5^{th}$ 2009 period (Fig. 2) by

comparing the model output (i.e., morphological features such as channel boundaries, islands, wet channel positions) to the channel morphology digitized using the 2009 aerial photos (see "Supplementary material" for a detailed description of model calibration). The hourly discharge series was used as upstream flow boundary condition. At the downstream end of the reach, a constant energy slope was fixed at 0.0047 m m$^{-1}$, equal to the local bed slope. The initial bed sediment grain size was set according to Tomasi (2009) results. The grain size distribution was defined using nine classes and was considered

homogeneous in the whole reach. Due to the lack of field estimates of bed load at the upstream end of the reach, we assumed the sediment recirculation option available in C-L (i.e., sediment input equals the output at the downstream end of the reach). The model factor called "Sediment Proportion Recirculated" (SPR) was assumed to be 1, which assumes that upstream sediment load being the same as at the downstream end of the reach (i.e., sediment transport equilibrium condition).





Vegetated areas (i.e., islands, recent and old terraces covered by arboreal vegetation) and channelization structures (i.e., bank
protection structures, groynes, and levees) were digitized combining 2003 aerial photos and LiDAR Digital Surface Model
(DSM, 2 m grid dimension, Table 1). The vegetation cover has been used as the model initial condition for vegetation
(maturity fixed to 1). The initial bed elevation was established using a 10 m cell DEM achieved by resampling the 2003
LiDAR DEM (bilinear interpolation, original cell dimension 2×2 m). The 10 m cell dimension was chosen to ensure a
reasonable computational time for long term scenario runs and also a spatial resolution higher than previous works (e.g. 25
m in Ziliani et al., 2013). The DEM was corrected in the wetted areas (about 8% of the total spatial domain) through the
application of the method proposed in Bertoldi et al. (2011) and forced to be "not erodible" in the areas occupied by both
channelization structures still effective in 2003 and undamaged structures built since the 19th century.

### 3.5 Flow-regime management strategies

In rivers historically regulated, water management may be oriented to restore flow regime close to the prior impact
conditions, typically aiming to reactivate physical processes linked to specific components of the flow regime (Wohl, 2011).
Nevertheless, existing priorities in uses of the water resources often limit the feasibility and the effectiveness of any flow
regime re-naturalization strategies, and in most cases the strategy is merely reduced to the definition of a minimum volumes
of water released for partial restoration goals. Olden et al. (2014) provided a systematic review of flood experiments to
evaluate globally the success of this practice in flow regime management. 113 flood experiments across 20 countries were
reviewed revealing that only 11% of case studies aimed directly to morphological effects and about 80% of the experiments
involved only low magnitude flow events. Major part of experimental flow releases focusses on a biological variable
(primarily fishes), aquatic organisms (Konrad et al., 2011) and vegetation reestablishment (Shafroth et al., 2010), rather than
abiotic factor (e.g., channel morphology) (Wohl et al., 2015b). The so-called "environmental flows" and "Green Hydro" are
concepts widely accepted even though they refer mainly to quantity, timing, duration, frequency and quality of water flows
releases, as required to sustain freshwater, estuarine and near-shore ecosystems, according to social interests (Acreman and
Ferguson, 2010; Rivaes et al., 2017).

Thus, there are few examples of flow regime recovery strategies which have been designed on a geomorphological basis or
rather planned to primarily achieve morphological targets. The Colorado River below the Glen Canyon Dams represents the
most relevant exception: several controlled floods (i.e., five High Flood Experiments) were released in the 1996-2013 period
to maintain and rehabilitate sandbars that occur in lateral flow separation eddies (Schmidt and Wilcock, 2008; Melis, 2011;
Mueller et al., 2014). Also the Trinity River (California, USA) represents an excellent case study, as a multiple objective
flow designed included morphological issue (Trinity Management Council, 2014). Still, no experiences are available on
morphological recovery or conservation of braided rivers, with the exception of the Lower Waitaki River (New Zealand),
even though no controlled floods have been used in that river (Hicks et al., 2003; Hicks et al., 2006; Environment Canterbury
Regional Council, 2015).



Controlled floods and vegetation removal actions are expensive and potentially forego other positive feedback effects. It is therefore worth to preliminary evaluate and test these kinds of actions taking into account the morphological effects of different channel-forming discharges (Surian et al., 2009). As emphasized by Rathburn et al. (2009), flow releases from dams must exceed a series of thresholds to be morphologically effective. Four increasing threshold discharges should be
identified in a morphological recovery/conservation plan, each able to activate a specific morphological process: (i) the mobilization of interstitial sediment essential for the hyporheic exchange maintenance; (ii) the mobilization of the streambed to maintain natural bedforms; (iii) the inundation of overbank units (i.e., berms, floodplains, terraces) to confine encroachment by xeric plants; and (iv) the lateral channel mobility that may promote the removal of senescent woody vegetation and create opportunities for seedlings to germinate and mature. Once each threshold discharge value or range is
quantified, the flow duration can be tuned within the limits imposed by the actual flow availability, assuming the natural flow regime as a reference.

In this work, the scenarios of flow regime management were defined referring to three flood threshold levels partially matching those proposed in Rathburn et al. (2009): (i) the in-channel full transport discharge assumed to be able to mobilize the interstitial sediment, (ii) the bankfull discharge able to maintain the natural bedforms; and (iii) the overbank discharge
able to affect the main lateral units by inundation. A field study conducted in the reach using painted sediments similar to (Mao and Surian., 2010; Mao et al., 2017) allowed to estimate that the first reference threshold is about 80 $m^3s^{-1}$ (RI < 1 year, full transport discharge able to mobilize sediments, irrespective of their size). The bankfull discharge was estimated using the calibrated C-L model and was approximated to the discharge filling the active channels and bars without overflowing onto the oldest island assumed to be morphologically equivalent to the recent fluvial terraces (Williams, 1978;
Pickup and Rieger, 1979; Surian et al., 2009). The bankfull discharge is about 500 $m^3s^{-1}$, and corresponds to a 2.5 years RI, coherently with other literature references (Leopold, 1994). Only the > 1,000 $m^3s^{-1}$ floods were identified to be able to completely inundate the oldest island and locally overflow into the recent fluvial terraces.

Four simulation scenarios, each 25 years long (2009-2034), were explored (Table 2). The first scenario (i.e., "baseline scenario", SC1) corresponds to the current condition characterized by a strongly altered flow regime. In this case, the
discharge series measured in 1995-2009 (hourly data at Belluno gauge station) has been repeated twice. Scenarios 2 (SC2) and 3 (SC3) were both set up using a flow regime strategy characterized by one CF per year. In SC2 the yearly CF had a constant value of 135 $m^3s^{-1}$ (RI ~ 1.08 years). This value has been calculated as the average of the maximum annual floods observed in 1995-2009, higher than the reference threshold discharge, i.e. the in-channel full transport discharge (80 $m^3s^{-1}$). In SC3, the yearly CF values were randomly selected above the threshold discharge using the natural streamflow pdf
estimated by the model (min value = 80 $m^3s^{-1}$; max value = 276 $m^3s^{-1}$, RI ~ 1.4 years). In this case, the average value of all the CF values was found to be equal to the SC2 yearly CF discharge, assuming values included in the [80 – 276] $m^3s^{-1}$ range. All the CFs had a fixed duration of 5 days, according to the re-naturalization maximization criteria. In SC2 and SC3, the cumulative likelihood to occur associated to the released peaks raised from 0.025 (actual observed altered regime) to 0.04, being the natural reference value 0.14. Scenario 4 (SC4) was planned to represent a different management strategy,



consisting in larger CF released by dams (constant value for one-day) only following the observation of notable channel narrowing. Specifically, we assumed 200 m as a threshold for average channel width, considering the evolutionary trajectory over the last 200 years and, in particular, the most intense narrowing that took place in the early 1990s (Fig. 3). Taking into account the channel width measurements conducted by a one-year step during the SC4 simulation, only two CFs were released, in 2020 and 2032 respectively. The released discharge was fixed equal to 600 m$^3$s$^{-1}$ (RI 5 years) ranging between

the second and the third reference threshold discharges discussed above, so that it was surely able to maintain the in-channel bedforms dynamics completely avoiding any hydraulic risk issues and damages in the overbank units (i.e., recent terraces), locally occupied by secondary roads and cultivated fields.

All the CFs have been released in November avoiding to overlap existing floods. All the scenarios made use of the same model setting achieved in the calibration phase, using the 2009 calibration run output data in raster format as initial boundary

conditions (i.e., bed elevation, not erodible hydraulic structures, vegetation cover, grain size bed sediment distribution). As reported in Table 2, all the scenarios required a cumulative volume to be released per year (i.e., per flood) considerably lower than the maximum stocked volume in the reservoirs existing upstream of the study reach (90.8 Mm$^3$). SC2 and SC3 required similar volumes of about 58 Mm$^3$ of water, corresponding to about 1.45x10$^3$ Mm$^3$ on the whole scenarios period. SC4 represents the cheapest scenario indeed because it needed 51.8 Mm$^3$ per year and 104 Mm$^3$ globally, one order of

magnitude lower than the other scenarios.

## 4. Results

### 4.1 Evolutionary trajectory of channel morphology

Using the available dataset on morphological changes of the Piave River (Comiti et al., 2011), we reconstructed and updated the channel adjustments up to 2009. The analysis focused on two sub-reaches, respectively upstream and downstream of San

Pietro in Campo where the river is naturally more confined (Fig. 1). The division in two sub-reaches was used to better describe the morphological adjustments over the 1800-2009 period. The average width trends are similar for both sub-reaches, and are characterized by four main adjustment phases (Fig. 3): (i) a first period (during the 19$^{th}$ century and the first half of the 20$^{th}$ century) dominated by braided pattern, channel width equal to about the 80% of alluvial plain width, negligible morphological changes and absence of a dominant process (i.e., channel widening or narrowing), (ii) a second

phase of adjustment with channel narrowing of about 60% from the 1950s to the early 1990s (whole reach channel width was 370 and 247, respectively in 1960 and 1991), interrupted by a large flood event in 1966 (RI ~ 200 years – (Comiti et al., 2011), which caused a temporary widespread channel expansion; (iii) a phase of channel widening during the 1990s (whole reach channel width was 342 m in 1999) mainly related to the 1993 flood event, characterized by 12 years RI (Comiti et al., 2011); and (iv) the most recent adjustment phase characterized by channel narrowing. Focusing on the last 20-25 years, after

the 2002 flood the river turned to a new phase (IV in Fig. 3) characterized by narrowing: channel width in 2009 (i.e. 241 m) was the lowest value observed in the study reach over the last 200 years. While during phase III widening was likely due to





the termination of in-channel gravel mining (Comiti et al., 2011), the most recent phase of narrowing (i.e., phase IV) was likely due to the absence of major floods (see also Fig. 2).

### 4.2 Flow regime alterations

The comparison between the frequency distribution of observed daily streamflows at Belluno cross section and the model-based estimate of the streamflow pdf under unregulated conditions (Fig. 4) shows the extent of the impact of regulation in the lower reaches of the Piave River. The mean and the mode of the streamflow distribution are significantly reduced by the anthropogenic exploitation of water resources (i.e., by-pass flows and diversions). Accordingly, the exceedance probability of moderate to high flows is significantly reduced under current regulated conditions. In particular, the probability to observe

discharges larger than 80 $m^3s^{-1}$ is reduced by about one order of magnitude (i.e., from 0.14 to 0.025). Such results are crucial for setting the flow-regime management scenario since (i) they show that a strategy aiming to improve the current flow-regime should be implemented, (ii) this strategy should compensate the expected low morphological dynamism of the river caused by the decreased occurrence of discharges able to mobilize sediments and produce significant morphological changes in the study reach.

It is worth to note that the hydrological model underestimates the frequency of the highest flows (i.e., discharges larger than 300 $m^3s^{-1}$) because all the non-linearities of the hydrologic response (e.g., the presence of different flow components such as surface runoff) are neglected in this version of the model (Basso et al., 2015). As a consequence, the probability associated to the highest flows in regulated conditions is larger than the corresponding value estimated by the stochastic model for the natural setting. This model limitation, however, does not bear any significant consequence for the analysis carried out in this

paper, provided that the frequency of such high flows is relatively low.

### 4.3 Calibration of the morphodynamic model

The results presented in Ziliani et al. (2013) and Coulthard et al. (2013) have been taken as a reference to achieve the C-L calibration. According to the results of the sensitivity analysis in Ziliani et al. (2013), lateral erosion rate and maximum erosion limit have been assumed as the most sensitive factors that required accurate tuning. The other factors (see Table 3),

including the main new parameters introduced in the C-L version, were tuned manually through a "trial-and-error" calibration strategy (n. 75 runs in total). Following the performance evaluation techniques used by Ziliani et al. (2013), the calibration was based on performance indices developed specifically for data available in raster format (Bates and De Roo, 2000; Horritt and Bates, 2001). The performance indices reported in Table 4 were calculated for all the calibration runs at the end of the simulation (2009), that is (i) the vegetation performance index ($F_{veg}$), (ii) the wet area performance index ($F_{wet}$)

and (iii) the active channel performance index ($F_c$). In addition, several planimetric features have been calculated including (i) average active channel width, (ii) equivalent wet area width ($L_w$), and (iii) mean braiding index (Egozi and Ashmore, 2008). The results (see Table 4, Fig. 5, Fig. S3 and S4 in the "Supplementary material" file) show a "very good performance" (performance class as defined in Henriksen et al., 2003; Allen et al., 2007) for both the vegetation cover ($F_{veg}$





69.7%) and the active channel area ($F_c$ 54.2%). Output values of the active channel width and braiding index values

confirmed these results. The difference between the real and modeled 2009 active channel width (6 m) is lower than input DEM cell size (10 m) and the modeled braiding index value (1.71) is very close to the real value (1.69). The model performance is poor only in reproducing the flowing channel position ($F_w$ 15.7 %) partially confirming results presented in Ziliani et al. (2013).

In order to integrate the morphological performance evaluations, we carried out an estimation of the mean annual bed load

sediment yield at the downstream end of the reach and along the whole reach. In the 2003-2009 period, the modeled average bed load sediment yield resulted of about 21.5 x $10^3$ $m^3yr^{-1}$. Modeled yield varies significantly along the reach (up to 30%) taking higher yearly values in the sub-reach upstream San Pietro in Campo. Significant differences exist between the maximum and minimum annual values. The 2006 minimum corresponds to an average annual sediment yield of about 260 $m^3yr^{-1}$ versus the 2008 maximum of about 53.3 x $10^3$ $m^3yr^{-1}$. Such sediment transport values agree with estimates for gravel-

bed rivers with similar characteristics to the Piave River reach (Martin and Church, 1995; Ham and Church, 2000; Nicholas, 2000; Liebault et al., 2008; Ziliani et al., 2013; Mao et al., 2017).

**4.4 Channel response to flow regime management strategies: scenario results**

Channel adjustments induced by all scenarios were assessed comparing every year (in February) the active channel width and the braiding intensity (BI) using the same techniques adopted in the calibration phase (Fig. 6). Channel width in

Scenarios 2-4 was almost always higher than in SC1 (the "baseline scenario"). On average, during the whole scenario period, SC2 and SC4 produced comparable channel widening of about 6.4% (~ 14 m), while at the end of the scenarios (2034), widening was about 9% (~ 25 m) and 13.5% (~ 38 m) in SC2 and SC4, respectively. The SC3 induced a slightly lower widening, about 5.4%, during the whole period, and 8.6% (~ 24 m) at the end of the simulated period. The maximum annual widening was observed in SC4 (~ 120 m in 2033), followed by SC3 (~ 77 m in 2020) and SC2 (~ 43 m in 2032 - Fig. 7).

Results suggest that the CFs scenarios (SC2-4) and the baseline scenario (SC1) provide similar long-term morphological trajectories characterized by alternate phases of widening and narrowing and notable changes in active channel width (width varies between 150 and 360 m). Figure 7 shows that each channel width oscillation takes place in about 6-7 years and it has 160 m amplitude in response to the alternation of periods characterized by different magnitude floods series: in 2011-2015 and 2022-2028 periods, during which seven floods > 400 $m^3s^{-1}$ (RI ~ 1.9 years) occur, channel width follows a quasi-steady

trend and is larger than 300 m. Instead, during the following periods (2017-2021 and 2029-2031) affected by lower magnitude floods (200 $m^3s^{-1}$ maximum peak value), channel width shows decreasing trajectories. Over the whole 25 years, SC1 provides a slightly decreasing trend (Fig. 7) that is not reversed in the other scenarios. In all the CFs scenarios the channel width trend assumes quasi-zero slope even if the channel width measured at the end of the simulations is about [8.6-13.5 %] higher than width in the baseline scenario.

The braiding index in Scenarios 2-4 was similar, or lower, than in SC1. SC4 resulted the scenario closest to the SC1, with a BI in-time averaged value equal to 2.78, only slightly lower than the SC1 (-1.5%). During SC2 we measured relative higher



differences in the BI value compared to SC1 (-7.3%, 0.21 BI unit – Fig. 7), although these differences are quite small. In terms of trajectory, braiding intensity shows a different behaviour in comparison to channel width, as there are no clear oscillation phases but one period (from 2009 to 2023) with a clear increasing trend, followed by a decreasing or quasi-steady

(SC1) period until the end of the simulation. There is a non-linear correlation between BI and flooding series magnitude or the CFs. In particular, SC1 is the only scenario that does not show a trend inversion after 2023, and SC2 scenario has a very anomalous trend showing a BI value steadily lower that the other CFs scenarios, while SC3 and SC4 show a good agreement in their BI trends.

## 5. Discussion

### 5.1 Geomorphic effectiveness of controlled floods

Comparing the future scenarios to the historical evolutionary trajectory (Fig. 3) several insights can be obtained despite the evident mismatch between the temporal frequency of the past and future channel width series (one value every 16.5 years in the 1805-1970 period and 6.5 years in 1970-2009 period; yearly values for the future series). It can be observed that: (i) the maximum channel widths reached during all of the four future scenarios (in the periods 2015-2016 and 2028-2029) are close

to the width in 1999, (ii) the minimum widths achieved in all future scenarios (2020 and 2032), are always below 185 m (with the exception of the first minimum during scenario 3) and are significantly lower than the historical minimum observed in 2009 (241 m), (iii) albeit with low confidence level we can state that the trajectory between 1991 and 2009 (phase III and IV described in Par. 4.1) seems to follow an oscillatory evolution with half the frequency of the oscillation modeled between 2009 and 2034, (iv) the correlation between the variation of channel width and the flow regime reproduced for the future is

not always straightforward in the past evolution. This point is exemplified by the rather major 2002 flood event which did not re-widen the river at the levels of 1999 (about 342 m), despite being relevant in terms of magnitude (13 years RI). Indeed, the following year (2003) the width was approximately 289 m, about 15% less than in 1999. This may suggest that the study reach, after a period (phases I and II in Fig. 3) of morphological instability characterized by a prevalent tendency to narrowing, has reached a new morphological equilibrium configuration characterized by a periodic oscillations of channel

width. Similar new equilibrium conditions, mainly controlled by flow regime (i.e. frequency and magnitude of formative discharges) and vegetation establishment, have been observed in the Tagliamento River (Ziliani and Surian, 2016).

The intercomparison of our four simulations shows that few high magnitude floods provide slightly better morphological recovery/conservation than small yearly floods, also at a significantly lower operational cost. Therefore, SC4 should be preferred to SC2 and SC3 from a pure morphodynamic point of view. Nevertheless, results suggest that (i) none of the CFs

scenarios are able to change significantly the long-term channel width and braiding intensity trends, (ii) CFs release have no significant morphological benefits and do not represent a solution for a morphological recovery in braided rivers that suffered such strong and historical impacts in terms of flow and sediment supply regimes. These results partially confirm the outcomes of Hicks et al. (2003) referring to the Waitaki River, a gravel-bed river with similar characteristics to the Piave





River. The authors state that, if a wider and more active channel is desired, an approach consisting in frequent release of

"channel maintenance floods" from dams should be pursued. Hicks et al. (2013) show that this kind of strategy may be unsuccessful and only multi-years high magnitude CFs can produce temporary stable widening channel condition.

The cost of CFs is probably smaller than those of alternative strategies focused on increasing sediment supply such as sediment augmentation, because flood releases commonly can be performed without redesign of reservoir structures. Nevertheless, reintroduction of flood flows implies "loss" of resource stocked for other purposes (e.g., hydroelectric

production, drinking or irrigation water supply). Another feasible way for sediment augmentation is the removal (at least in part) of non-strategic bank protections along the reach. However, as suggested by Picco et al. (2016), this kind of strategy should be preventively assessed since these structures are still viewed by local populations as necessary to protect riparian woodlands that are highly appreciated for recreation and timber production.

Overall, this work gives useful insights for the Piave River management and, in general, for braided rivers heavily impacted

in flow and sediment regimes: (i) none of the controlled flood strategies that was tested is able to significantly change the on-going morphological evolution; (ii) the baseline scenario, without controlled flood releases (i.e., the no action strategy), provides morphological evolution trajectory similar to that induced by the controlled floods release scenarios.

## 5.2 Assessment of CAESAR-LISFLOOD performance

In Ziliani et al. (2013) the authors concluded that the main factors causing the morphological poor response of CAESAR are

(i) the DEM cell size, as pointed out also in others works (Doeschl-Wilson and Ashmore, 2005; Doeschl et al., 2006; Nicholas and Quine, 2007), (ii) the quality of data (i.e., lack of wet channel topography) and (iii) the low flow periods removal, and therefore the cut off of the consequent morphological "gardening" phenomena (Ziliani et al., 2013). The combination of these factors produced a smoother and simpler braided morphology. The Piave case study represents an effort to achieve a better performance by (i) the flow routines refinement included in the LISFLOOD-FP module (one of the

most recent and advanced Reduced Complexity Hydraulic Model scheme), (ii) the adoption of input data of higher quality (higher resolution DEM, bathymetry and hourly boundary conditions) and (iii) the code conversion in parallel programming methods. The results lead to an overall improvement of the model performance considering (i) the good channel width performance in the calibration phase, (ii) the excellent braiding complexity reproduction, including the pioneer and complex islands dynamics, both in the calibration and in long-term simulations, (iii) the reasonable estimation of bedload transport,

and (iv) the adequate computation speed, close to the expectations (i.e., about 10 days of computation for 25 years of hourly series).

The suitability of the RCMs application for the investigation of river dynamics has been discussed in several previous studies (Doeschl-Wilson and Ashmore, 2005; Brasington and Richards, 2007; Nicholas and Quine, 2007; Murray, 2007; Nicholas, 2012, 2013b; Ziliani et al 2013; Ziliani and Surian, 2016). A general conclusion of these works is that RCMs may provide

morphological responses both unrealistic and highly sensitive to model grid resolution. These problems are commonly interpreted as a direct consequence of both the adoption of flow routing schemes that neglect the momentum conservation



and the use of local bed slopes for the bedload transport calculation (e.g., through the application of the uniform flow approximation).

The C-L model may be considered an useful tool in the search of an effective combination of simplicity and physical realism
in the context of reduced-complexity modelling, overcoming some of the previous hydrodynamic simplification issues. The encouraging results achieved in this case study seem to justify the effort faced in such further development of this RCM. Although the physical realism of flow and morphodynamic rules can remain unsolved at smaller scales (i.e., scales lower than DEM cell dimension), the improvement of the C-L model response at reach scales compared to the older CAESAR model is evident. Although the reduced-complexity modelling approach probably will not provide insights into some of the
reductionist key questions currently faced by hydraulic engineers and fluvial geomorphologists, the model can provide useful insights for management. Specifically, insights about their macro morphological features (e.g. average channel width, braiding intensity) and adjustments (e.g. prediction of future evolutionary trajectory) of braided rivers.

The inherent limitation in reduced-complexity modelling approach does not preclude the adoption of RCMs where the aim is to represent meso-scale system behaviour, rather than to make reductionist predictions that are theoretically more accurate in
quantitative terms. Reproducing the morphodynamic processes at each scale required necessarily some forms of simplification, regardless of the level of complexity of the model adopted, and in CFD models as well. Nevertheless, this cannot be considered a convincing reason to necessarily calling into question whether explanations of river behaviour based on this kind of model have any application in the real world. In light of the results presented in this work and of the limitations faced anyway by the reductionist alternative approaches (Williams et al., 2016), we believe that RCMs, and C-L
model specifically, remain an attractive option for simulating river evolution over historical time periods and future scenarios, that is at the scale of interest for river management.

This work presents another case study in which an RCM has given realistic outputs in a large gravel-bed river, especially in terms of evolutionary trajectories. The suitability in reproducing macro morphological features and meso-scale processes should not be questioned any longer (Nicholas, 2013b). The capability to model small-scale phenomena remains open for
RCMs as for all CFDs that try to reproduce phenomena deeply influenced by initial and boundary conditions, for which a data gap persists for future scenario application in natural contexts where the addition of modelling details does not guarantee a significant reduction of the overall uncertainty associated to the model results.

## 6. Conclusions

Hydrological and morphodynamic models have been applied to assess the long-term geomorphic effectiveness of controlled
floods strategies. The simulated future scenarios (with a duration of 25 years) show that: (i) none of the CFs strategies can provide significant long-term morphological benefits and is able to reverse the ongoing channel width trend; (ii) few high magnitude floods (i.e. SC4) provide slightly better morphological recovery than small yearly floods (i.e. SC2 and SC3), also at a significantly lower operational cost (the cumulative volume released in SC4 is an order of magnitude lower than in SC2





and SC3). These results suggest that this kind of strategy does not represent a solution for morphological recovery in braided

rivers strongly and historically impacted.

The study confirms the suitability of the RCMs for modelling long-term future scenarios at spatial and temporal scales still hardly supported by 2D-3D CFD morphodynamic models. From a morphological point of view, the C-L model has proven to be able to reproduce the channel width variation, to preserve the morphological braiding complexity, including the vegetation dynamics, and to estimate reasonably the average bed load sediment yield. The model performance assessment

shows significant improvements of C-L model in comparison to previous CAESAR model version (Ziliani et al. 2013). The application of this RCM does not provide insights into the spatial and temporal scales of interest for a traditional reductionist approach (e.g., single branch and bar dynamics, local bank erosion) however it provides useful indications for management of braided rivers at meso-scales.

**Code availability**

The CAESAR-LISFLOOD model code is freely available at: https://sourceforge.net/projects/caesar-lisflood/

**Data availability**

LiDAR data and aerial photos (2003) are available upon request at the Autorità di Bacino delle Alpi Orientali.
Hydrological data are available upon request at the Environmental Regional Agency (ARPA Veneto).
2009 aerial photos and cross sections are freely available by contacting the authors.

**Supplement**

The supplementary material related to this article is available online at: http://researchdata.cab.unipd.it/id/eprint/157

**Author contribution**

LZ and NS designed the research. LZ performed most of the analyses. GB aided as expert in hydrology and hydrological modelling. LM provided guidance on sediment transport. All authors jointly contributed to the discussion and interpretation

of the data. The paper was prepared by LZ, with contributions from NS, GB and LM. NS managed and coordinated research activities.

**Competing interests**

The authors declare that they have no conflict of interest.





**Acknowledgements**

This research was supported by funds from the University of Padova: strategic project "GEO-RISKS" and DOR funds; we would like to thank the Autorità di Bacino delle Alpi Orientali for providing the LiDAR data.

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



**Figure 1. (a) Study reach location. (b) Water infrastructure system in the Piave Basin**

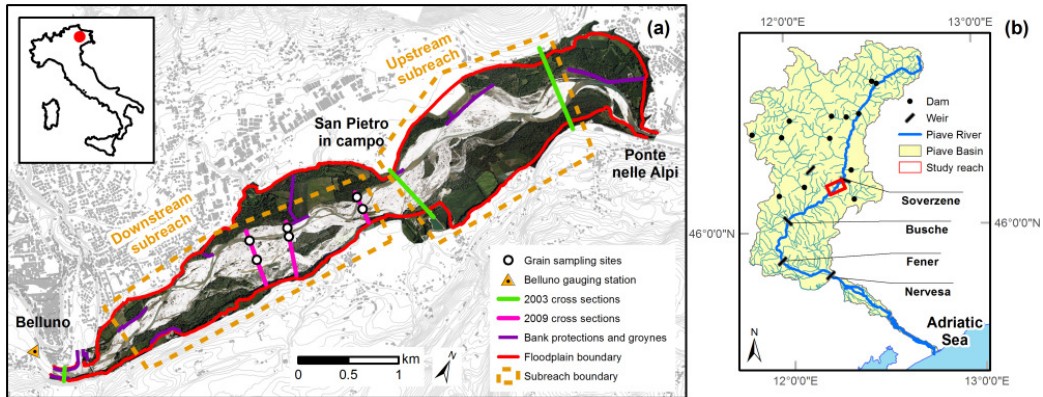





**Figure 2. (a) June 1st 1995 - December 31th 2009 hourly discharge series used for the scenarios runs; (b) Hourly discharge series measured at Belluno gauge station used for the calibration run (July 5th 2003 to August 5th 2009)**

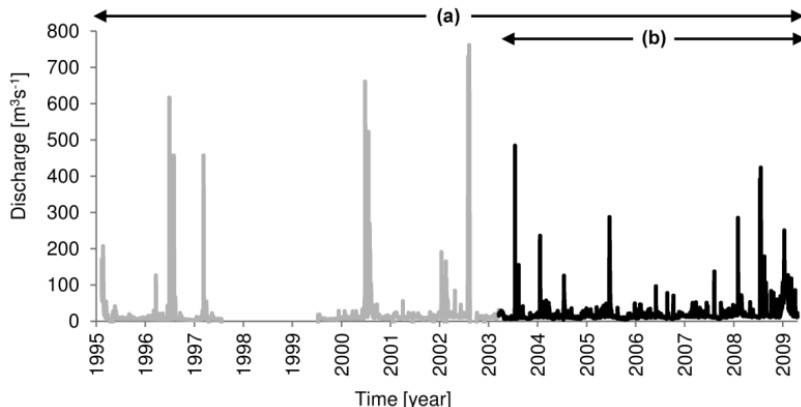



**Figure 3. (a) Changes in channel width over the period 1805–2009; (b) changes in channel width over the period 1960–2009**

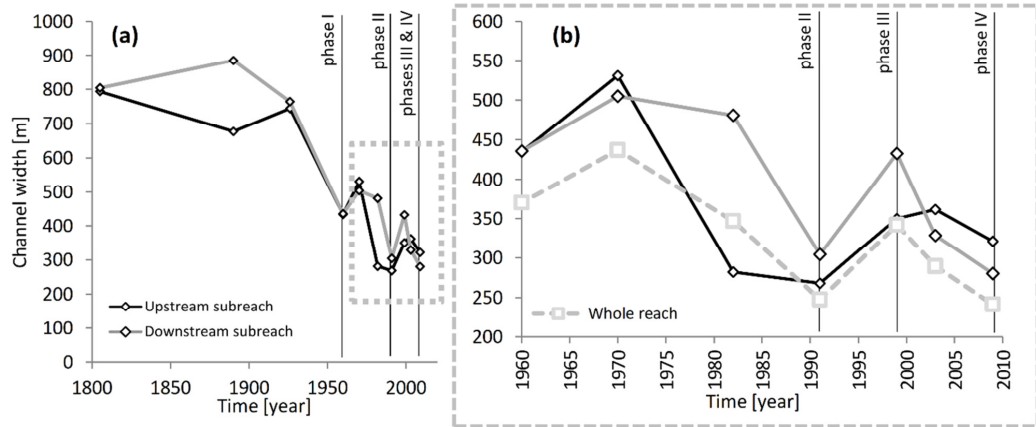





**Figure 4. Effect of the anthropogenic regulations in the Piave River at Belluno cross section. Comparison between the pdf p(Q) (a) and the cumulative distribution D(Q) (b) of observed streamflow in the period 1995 – 2009 and the natural streamflow pdf and D(Q) predicted by the model developed by Botter (2010)**

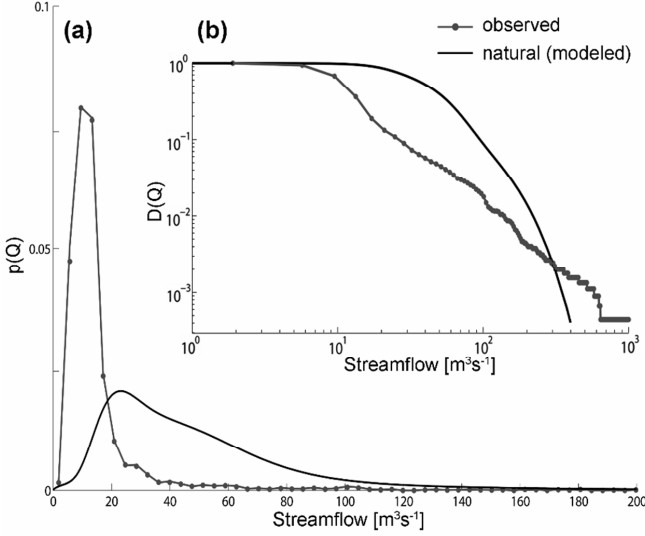




**Figure 5. Model performance assessment at the end of the calibration runs (mid frame – see Supplementary material file for the others frames): (a) wet area (b) vegetated area and (c) active channel digitalized using 5th August 2009 aerial photos; (d) overlay between modeled and observed flowing channel; (e) overlay between modeled and vegetated area; (f) overlay between modeled and observed active channel; (g) wet area performance index calculation; (h) vegetated area performance index calculation; (i) active channel performance index calculation**

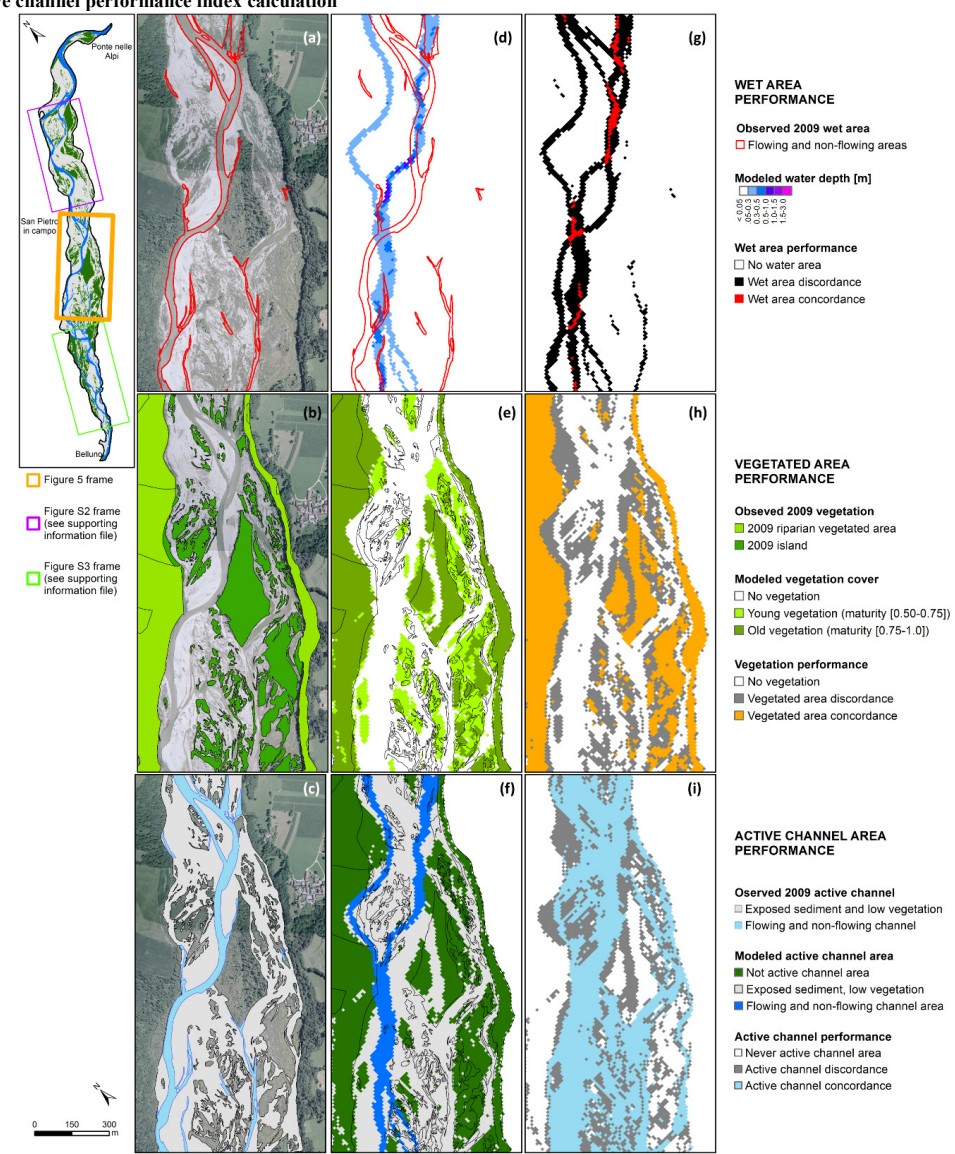




**Figure 6. (a) 2009 real channel, (b) 2009 modeled channel used as starting point for the scenario runs; (c-f) Scenarios 1-4 results at the end of the runs (2034)**





**Figure 7. Scenario results expressed in terms of channel width (a) and braiding index (b) variations, coupled to the simulated upstream inflow series with controlled flood releases: (c) scenario 1 and 2, (d) scenario 3, (e) scenario 4**

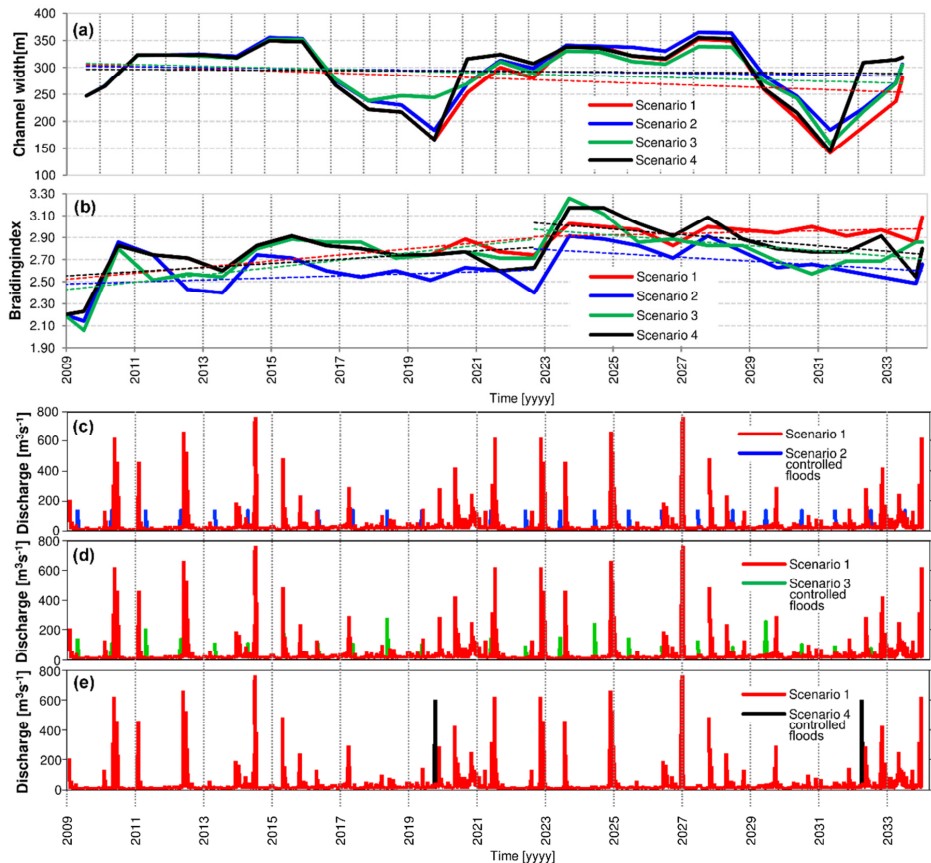





**Table 1. Summary of the data used for the historical adjustment analysis and modelling activity**

| Type | Source | Data | Main characteristics | Location |
|------|--------|------|----------------------|----------|
| Aerial photos | Autorità di Bacino delle Alpi Orientali | July 5th 2003 | Resolution: 30 cm | The whole study reach |
| | Department of Geosciences, University of Padova | August 5th 2009 | Resolution: 15 cm | The whole study reach |
| Cross section survey | Autorità di Bacino delle Alpi Orientali | 2003 | DGPS topographic survey. | See Fig. 1 |
| | Department of Geosciences, University of Padova | March 2009 | RTK DGPS topographic survey | See Fig. 1 |
| DEM / DSM | Autorità di Bacino delle Alpi Orientali | July 5th 2003 | LiDAR DEM / DSM, grid dimension: 2 m | The whole study reach |
| Bed grain size measurements | [Tomasi, 2009] | July 2008 | 6 samples. Volumetric method. Sampled depth 0.5 m | See Fig. 1 |





**Table 2. Main hydrological characteristics of the four scenarios**

| Scenario | Controlled flood releases frequency | Maximum controlled flood peaks [m³s⁻¹] | Controlled flood peaks range [m³s⁻¹] | Recurrence interval maximum controlled flood [year] | Controlled flood duration [days] | Cumulative volume released per flood [Mm³] | Cumulative volume released per scenario [Mm³] |
|---|---|---|---|---|---|---|---|
| SC1 [baseline scenario] | No releases | No releases | No releases | - | - | 0 | 0 |
| SC2 | One per year | 135 | Constant | 1.08 | 5 | 58.32 | $1.46 \times 10^3$ |
| SC3 | One per year | 276 | [80 - 276] | 1.4 | 5 | 58.18 | $1.45 \times 10^3$ |
| SC4 | In case of average channel width narrowing under 200 m (2 times in 34 years) | 600 | Constant | 5 | 1 | 51.84 | $1.04 \times 10^2$ |





**Table 3. Description of the CAESAR-LISFLOOD model calibrated factors**

| Factor [a] | Investigated range | | Calibration setting |
|---|---|---|---|
| | Min | Max | |
| Lateral erosion rate [-] | 0.002 | 600 | 30 |
| Maximum erosion limit [m] | 0.001 | 0.075 | 0.01 |
| Active layer thickness [m] | 0.004 | 0.28 | 0.04 |
| Number of passes for edge smoothing filter [-] | 30 | 200 | 150 |
| Water depth above which erosion can happen [m] | 0.01 | 0.15 | 0.15 |
| Bed load solid transport formula [b] | 0 | 1 | 1 |
| Vegetation critic shear [Nm$^{-2}$] | 0.7 | 180 | 0.9 |
| Vegetation maturity [year] | 0.06 | 20 | 4 |
| Courant number [-] | 0.1 | 0.7 | 0.2 |
| Input/output difference allowed [m$^3$s$^{-1}$] | 1 | 5 | 5 |
| In-channel lateral erosion rate [Nm$^{-2}$] | 1 | 30 | 10 |
| Slope for edge cells [d] [-] | - | - | 0.005 |
| Sediment proportion recirculated [d] [-] | - | - | 1 |

[a] All factors are configurable using the graphical user interface

[b] 0 - Einstein (1950) formula; 1 - Wilcock and Crowe (2003) formula

[c] Factor not included in the old CAESAR model version

[d] Factor not calibrated



**Table 4. Results of CAESAR-LISFLOOD calibration**

|  | Observed active channel 2009 [rasterized data - 10x10 m] | CAESAR-LISFLOOD result |
|---|---|---|
| Active channel width [m] | 241 | 247 |
| Active channel width change in 2003-2009 period [m] | - 35 | - 29 |
| Wet are width [m] | 51 | 66 |
| Braiding Index [channel counted] | 1.69 | 1.71 |
| Performance active channel [%] |  | 54.2 % |
| Performance vegetated area [%] |  | 69.7% |
| Performance wet area [%] |  | 15.7% |
