# Peer review of "Assessment of the geomorphic effectiveness of controlled floods in a braided river using a reduced-complexity numerical model"

_Hydrology and Earth System Sciences, 2019_

## Referee Comment (RC1) · Anonymous Referee #1 · 1 Dec 2019

The authors apply a 2D reduced complexity morphodynamic model to a 7 km-long reach of the braided Piave River. Their goal is to establish that the model credibly represents changes in channel morphology and to ask whether artificial floods might change the future morphology. The applied interest is maintaining channel width and braiding complexity on a river that is progressively narrowing and simplifying due to water management.

The paper is generally well organized and written clearly, with an appropriate amount of documentation. There are about 25-30 small examples in which a careful copy editor is needed to correct English phrasing. These errors rarely produce ambiguity, but should

be corrected. I did not have time to mark them myself.

The authors do a good job explaining the model and testing its suitability, They approach the difficult issue of matching model prediction and reality with care. My comments are intended to suggest additional means to explain, evaluate, and justify the model results. I think they should be addressed by the authors, although they do not all need to be acted on.

(1) My first look at the test hydrographs (Figure 7.c, d, e) suggested to me that little difference in predicted channel morphology should be anticipated. The controlled floods in Scenarios 2 and 3 are too small and the controlled floods in Scenario 4 are too infrequent (and no larger than natural floods). Presumably the flood scenarios chosen are as large as can be released given the water infrastructure in the basin. Hence, the (not too surprising) result is that the modest or infrequent floods that are feasible are not sufficient to produce significant changes in the forecast channel morphology.

(2) An interesting way to present the results would be in terms of 'limits of prediction'. That is, conduct multiple runs driven by small changes in parameter or initial conditions or in the sequencing of floods, in order to show how variable the results would be given uncertainty of the input. I would guess that the range of predicted width and braiding index (Figure 7.a, b) would comfortably encompass the predictions from the different scenarios, indicating that the model is unable to demonstrate that the available floods would produce different morphologies.

(3) I do wonder whether the model is able to predict larger widths. Is the model capable of predicting a width of, say, 430 m, as observed in 1970? I realize that flows up to 1970 are outside of the calibration range, but I would be concerned about whether the apparently firm upper bound on width of 350 m (Figure 7.a) is somehow an artifact of the model.

(4) The test for sediment transport rate is quite weak: the authors find that the computed transport rates are within typical range for such gravel-bed braided rivers. A
more sensitive test would be to evaluate how the bed grain size changes over time. You specify an initial grain size - does that grain size shift dramatically over the course of the model run?

(5) River are a combination of sediment-feed and sediment-recirculating systems. I suspect that the model results are sensitive to this choice. The problem, of course, is specifying an upstream sediment boundary condition. I would be interested in learning how an increase in sediment supply changes the predicted channel morphology. Perhaps that is beyond the scope of the paper, although the authors do mention that sediment mining was practiced and then halted. Model runs with and without substantial sediment removals would certainly be interesting!

———————————————

---

## Referee Comment (RC2) · Tom Coulthard (Referee) · 22 Jan 2020

This is a really neat study looking at how a reduced complexity morphodynamic model can be used to investigate how the hydrological pertubation of flows within a managed braided river affect the morphology. Ultimately, it shows how controlled releases of larger flows have some but not significant impacts on channel widths and depths within the reach studied.

In doing so it also provides an excellent opportunity to evaluate/test a morphodynamic model (which is a complex and far from straightforward exercise) demonstrating how such methods can be used as management tools in such envronments to answer questions about hydro-geomorphic interactions.

The paper is well written, produced and structured. As per R1, there are several minor grammatical/typo mistakes that can be picked up in proof reading if the paper progresses. Some specific comments and suggestions for further literature that has not been cited (some has only just come out) are provided below.

65 Impellent?

70 A reference to Larsen et al https://doi.org/10.1002/2014EO320001 might be useful in the RCM description section here.

230-236 - felt a bit clunky and repetetive - might be worth having a closer look at this section. Also there have been a series of CL papers and studies since 2013 that might be useful for the paper to cite here as well. THere are others but two more recents ones CL Sensitivity analysis paper: https://www.geosci-model-dev.net/11/4873/2018/ Calibrating valley floor re-working in CL. Feeney et al., 2020 https://doi.org/10.1002/esp.4804

237 Section 3.4. The two references above are also really relevant for this section - there have been more thoughts and studies on validation/calibration methods. Both compliment what you are doing here I think.

---

## Author Response (AR1)

"Assessment of the geomorphic effectiveness of controlled floods in a braided river using a reduced-complexity numerical model" Ziliani et al.

5

**General Response to reviewers**

Dear Editor,

We thank the reviewers for their thoughtful and useful comments. All comments are constructive, they are very useful to clarify and improve some specific aspects of the manuscript. In this rebuttal we address all the reviewer comments: detailed responses are provided in bold below.

Both reviewers pointed out some problems with English: copy editing was carried by a professional mother tongue.

15 Regards

Nicola Surian

April 30, 2020

20

**Response to reviewer 1 (Anonymous)**

25 The authors apply a 2D reduced complexity morphodynamic model to a 7 km-long reach of the braided Piave River. Their goal is to establish that the model credibly represents changes in channel morphology and to ask whether artificial floods might change the future morphology.

The applied interest is maintaining channel width and braiding complexity on a river that is progressively narrowing and simplifying due to water management.

30 The paper is generally well organized and written clearly, with an appropriate amount of documentation. There are about 25-30 small examples in which a careful copy editor is needed to correct English phrasing. These errors rarely produce ambiguity, but should be corrected. I did not have time to mark them myself.

The authors do a good job explaining the model and testing its suitability. They approach the difficult issue of matching model prediction and reality with care. My comments are intended to suggest additional means to explain, evaluate, and justify the model results. I think they should be addressed by the authors, although they do not all need to be acted on.

(1) My first look at the test hydrographs (Figure 7.c, d, e) suggested to me that little difference in predicted channel morphology should be anticipated. The controlled floods in Scenarios 2 and 3 are too small and the controlled floods in Scenario 4 are too infrequent (and no larger than natural floods). Presumably the flood scenarios chosen are as large as can be released given the water infrastructure in the basin. Hence, the (not too surprising) result is that the modest or infrequent floods that are feasible are not sufficient to produce significant changes in the forecast channel morphology.

40

45

50

55

Thanks for this comment. As suggested by the reviewer, we did not expect major changes in channel morphology but, considering the controlled floods used in the three Scenarios, some significant changes could be expected. As already explained in the manuscript, the controlled floods are feasible, that is taking into account the water infrastructure in the basin. On the other hand, it is worth noting that the floods in Scenarios 2 and 3 are not so small (these are formative discharges and are released for 5 days) (Table 2). We agree that controlled floods in Scenario 4 are infrequent, but those floods are quite large (recurrence interval = 5 years) and released for 1 day (Table 2). Therefore, some effects on channel morphology (i.e. some geomorphic recovery) could be expected. The results show that any significant recovery took place. A main outcome of this paper, that could not be anticipated when we started this research, is that controlled floods may

have any significant effects in a strongly regulated river, specifically if formative discharges have been strongly altered. In terms of changes in the manuscript, we carried out small changes in the Discussion (Section 5.1; L 454-456) and in the Conclusions (L 525-526).

(2) An interesting way to present the results would be in terms of 'limits of prediction'. That is, conduct multiple runs driven by small changes in parameter or initial conditions or in the sequencing of floods, in order to show how variable the results would be given uncertainty of the input. I would guess that the range of predicted width and braiding index (Figure 7.a, b) would comfortably encompass the predictions from the different scenarios, indicating that the model is unable to demonstrate that the available floods would produce different morphologies.

We agree with the reviewer, a sensitivity analysis would be needed for better assessment of the results and associated uncertainty. On the other hand, since a comprehensive testing of the model is very complex (this is also pointed out by reviewer 2 "...to evaluate/test a morphodynamic model (which is a complex and far from straightforward exercise)...") and we could rely upon a sensitivity analysis that we carried out in a similar river (Ziliani et al, 2013, JGR), the sensitivity analysis was out of the scopes of this work. That said, we carried out a calibration (Section 4.3) which shows a very good performance of the model, specifically as for estimate of channel width and braiding index.

(3) I do wonder whether the model is able to predict larger widths. Is the model capable of predicting a width of, say, 430 m, as observed in 1970? I realize that flows up to 1970 are outside of the calibration range, but I would be concerned about whether the apparently firm upper bound on width of 350 m (Figure 7.a) is somehow an artifact of the model.

Thanks for this comment. We think that the model should be able to simulate larger widths and that the lack of very large floods in the scenarios prevented further

3

60

65

70

75

85

widening (i.e. channel widths larger than 350 m). To confirm this hypothesis, we run a new simulation which includes a large flood (peak value =  $1600 \text{ m}^3\text{s}^{-1}$ ; RI = 30 years). This flood produced a remarkable widening being average channel width 355 m and 430 m, respectively before and after the flood (Fig. 1).

- (4) The test for sediment transport rate is quite weak: the authors find that the computed transport rates are within typical range for such gravel-bed braided rivers. A more sensitive test would be to evaluate how the bed grain size changes over time. You specify an initial grain size - does that grain size shift dramatically over the course of the model run?
- 105 Thanks for pointing out this. We analyzed grain size changes, specifically we compare D50 of bed sediment at the beginning (i.e. 24.9 mm) and at the end of each

Scenario. The D50 changes are small, since D50 is 22.7, 24.3, 23.6 and 23.1 respectively at the end of Scenario 1, 2, 3 and 4. In the revised manuscript this information was included in Section 5.2 ("Assessment of CAESAR-LISFLOOD performance"; L 486-487).

110

115

- (5) River are a combination of sediment-feed and sediment-recirculating systems. I suspect that the model results are sensitive to this choice. The problem, of course, is specifying an upstream sediment boundary condition. I would be interested in learning how an increase in sediment supply changes the predicted channel morphology. Perhaps that is beyond the scope of the paper, although the authors do mention that sediment mining was practiced and then halted. Model runs with and without substantial sediment removals would certainly be interesting!
- We agree, in general for such dynamic systems it is crucial to take into account also possible changes in sediment supply. In this case study sediment supply was kept constant for two reasons. First, including changes in sediment supply would imply to carry out several other Scenarios, adding complexity to the modelling and to the overall work. Second, and most importantly, there is evidence that the study sector is not undergoing significant vertical changes (i.e. incision or aggradation) over the recent period (see lines 123-124 of the manuscript). Besides, it is not likely that sediment mining will be carried out in the study sector in the near future. For such reasons, the sediment recirculation option of the model (i.e. sediment transport equilibrium condition) was adopted for this study.
- 130

**Response to reviewer 2 (Tom Coulthard)**

This is a really neat study looking at how a reduced complexity morphodynamic model can be used to investigate how the hydrological perturbation of flows within a managed braided river

affect the morphology. Ultimately, it shows how controlled releases of larger flows have some but not significant impacts on channel widths and depths within the reach studied.

In doing so it also provides an excellent opportunity to evaluate/test a morphodynamic model (which is a complex and far from straightforward exercise) demonstrating how such methods 140 can be used as management tools in such environment to answer questions about hydrogeomorphic interactions.

The paper is well written, produced and structured. As per R1, there are several minor grammatical/typo mistakes that can be picked up in proof reading if the paper progresses.

Some specific comments and suggestions for further literature that has not been cited (some has only just come out) are provided below.

- 65 Impellent?
   We changed this adjective
- 70 A reference to Larsen et al https://doi.org/10.1002/2014EO320001 might be useful in the RCM description section here.
   The suggested paper was cited, as it properly emphasizes the exploratory purpose of RCMs application in fluvial geomorphology as well as in other complex Earth

and environmental systems.

- 230-236 felt a bit clunky and repetetive might be worth having a closer look at this section. Also there have been a series of CL papers and studies since 2013 that might be useful for the paper to cite here as well. There are others but two more recent ones CL Sensitivity analysis paper: <a href="https://www.geosci-modeldev.net/11/4873/2018/Calibrating\_valley\_floor\_re-working\_in\_CL\_Feeney\_et\_al., 2020">https://www.geosci-modeldev.net/11/4873/2018/Calibrating\_valley\_floor\_re-working\_in\_CL\_Feeney\_et\_al., 2020</a>
   https://doi.org/10.1002/esp.4804.
   Thanks for pointing out these two papers which represent the most recent applications of the CL model. Skinner et al. (2018) is focused on a global sensitivity analysis applied to CL model (catchment mode application) used as Landscape Evolution Model. Feeney et al. (2020) focuses on the effectiveness of CL model in
- reproducing historical channel lateral erosion and modelling floodplain turnover (in single thread sinuous o meandering reaches, locally wandering).

We revised the last part of section 3.3. (L 224-228), including the suggested papers and also Coulthard and Van De Wiel (2017).

• 237 Section 3.4. The two references above are also really relevant for this section there have been more thoughts and studies on validation/calibration methods. Both compliment what you are doing here I think.

We agree with the Reviewer. Both papers were cited in the revised version of the manuscript, since they are complementary to our work (Skinner et al., 2018, see L 226; Feeney et al., 2020, see L 227 and L 240).

7

**Assessment of the geomorphic effectiveness of controlled floods in a braided river using a reduced-complexity numerical model**

Luca Ziliani1, Nicola Surian1\*, Gianluca Botter2, Luca Mao3

2 Department of Civil, Environmental and Architectural Engineering, University of Padova, Italy

3 School of Geography, University of Lincoln, United Kingdom

185 Correspondence to: Nicola Surian (nicola.surian@unipd.it)

Abstract. Most Alpine rivers have undergone strong alteration of significant alterations in flow and sediment regimes. These alterations have notable effects on river morphology and ecology. One option to mitigate such effects is the-flow regime management, specifically bythrough the re-introductionreintroduction of channel-forming discharges. The aim of this work is to assess the morphological changes induced in the Piave River (Italy) due to by two different distinct controlled flood strategies,

- 190 the first characterized by a single artificial flood per year and the second by higher magnitude, but less frequent, floods. The This work was carried outinvolved applying a 2D reduced-complexity morphodynamic model (CAESAR-LISFLOOD) to a 7-km-long reach, characterized by a braided pattern and highly regulated discharges. The numericalNumerical modelling allowed the assessment of morphological changes for four long-term scenarios (2009-2034). The scenarios were defined taking into accountconsidering the current flow regime and the natural regime, which was estimated by a stochastic physically-based
- 195 hydrologic model. Changes in channel morphology were assessed by measuring active channel width and braiding intensity. ComparingA comparison of controlled flood scenarios to a baseline scenario (i.e., no controlled floods) it turned outshowed that artificial floods had small effectslittle effect on channel morphology. The highestMore channel widening (13.5%) was produced byresulted from the release strategy with higherhigh magnitude floods, whileflood strategy than from the application of the other strategies produced lower wideningstrategy (8.6%). Negligible change was observed in terms of braiding intensity.
- 200 Results pointed outThe results indicate that controlled floods maydo not represent an effective solution for morphological recovery in braided rivers with strongly impacted in their flow and sediment regimes.

**1 Introduction**

Human activities in riverine areas (i.e., river damming, river and engineering, gravel mining, land-use change in the drainage basin) have historically led to notable changes in the flow regimeregimes (Gore and Petts, 1989; Poff et al., 1997; Magilligan and Nislow, 2005; Poff et al., 2007; Zolezzi et al., 2011; Magilligan et al., 2013; Ferrazzi and Botter, 2019), along withand in ecological (Collier, 2002; Céréghino et al., 2004; Paetzold et al., 2008; McDonald et al., 2010; Overeem et al., 2013; Espa et al., 2015) and geomorphic functioning of river systems (Hicks et al., 2003; Petts and Gurnell, 2005; Melis, 2011; Ziliani and

8

<sup>1 Department of Geosciences, University of Padova, Italy

Surian, 2012; Magilligan et al., 2013; Mueller et al., 2014; Lobera et al., 2016). Nowadays, damDam construction is now considered a viable strategy to support the increasingmeet energy and water demands due to climate change and population 210 growth (World Bank, 2009; Lehner et al., 2011). As outlined by Overeem et al. (2013), large reservoirs with a volumevolumes greater than 0.5 km3 intercept globally more than 40% of river discharge and ~26% of the sediments transported by the rivers, reducing the global sediment delivery to oceans, and commonly leading to coastal erosion.

Several metrics have been developed to assess both the magnitude and temporal trend of trends in alterations of in river flow regime in riversregimes induced by hydraulic infrastructures infrastructure (Richter et al., 1996; Richter et al., 1997; Martínez 215 Santa-María et al., 2008; Yin et al., 2015), and). As well, extensive sediment monitoring efforts orand sediment budget

- estimations have quantified sediment flux alterations (Surian and Cisotto, 2007; Schmidt and Wilcock, 2008; Melis, 2011; Trinity Management Council, 2014; Espa et al., 2015). Several studies have documented the hydrologic impacts following the of extensive realization of dam systems, in particular in the Alpine region over the 20th century (Botter et al., 2010; Comiti, 2012; Bocchiola and Rosso, 2014). Overall, flow regime alteration has implied significant changes in flow magnitude,
- frequency, timing and duration, and thermo-peaking phenomena (Gore and Petts, 1989; Frutiger, 2004; Zolezzi et al., 2009; Zolezzi et al., 2011). Impacts The effects of damming on sediment fluxes flux have been assessed onin reaches directly impacted by dammingaffected (Graf, 1980; Williams and Wolman, 1984; Gaeuman et al., 2005; Petts and Gurnell, 2005; Schmidt and Wilcock, 2008; Grant, 2012)), as well as in lowland gravel-bed rivers affected by cascading connected reservoirs and other human disturbances at the basin scale (Rinaldi and Simon, 1998; Surian and Rinaldi, 2003; Bilotta and Brazier, 2008; Surian, et al., 2009; Zawiejska and Wyżga, 2010; Ziliani and Surian, 2012; Scorpio et al., 2015). 225

- Overall, itlt is widely acknowledged that a reduced sediment flux due to dam construction or sediment supply alteration at the basin scale (e.g. due to afforestation, torrent-control works) produces channel changes (, namely, narrowing, incision, incisions, reduced braiding intensity reduction), and coarsening of bed sediment. Since the 1970s, growingThere has been increasing attention was paid tosince the 1970s on the environmental effects of large dams (Turner, 1971; Vörösmarty et al., 2003).
- 230 Different river management strategies have been adopted to address dam-related impacts using structural-or, operational strategies (e.g., Kondolf et al., 2014), or process-based approaches oriented toward restoring to restore water and sediment fluxes (Wohl et al., 2015a). Flow releases from dams have been eventually regulated to reproduce some aspectaspects of the natural regimes (flow and sediment), via increasing or recovering seasonal baseflow increase or recovery (McKinney et al., 2001; Sabaton et al., 2008), control incontrolling the timing and recession rates of releases (Rood et al., 2003; Shafroth et al.,
- 235 2010), artificial gravel augmentation or sediment bypassing (McManamay et al., 2013; Kondolf et al., 2014), flood releases (Collier, 2002; Dyer and Thoms, 2006) or high-flowand experimental high-flow releases (Melis, 2011; Olden et al., 2014). In other cases, the management strategies have focused directly on recovering morphological features recovery through dam removalby removing dams (Poulos et al., 2014; O'Connor et al., 2015), or mechanicalby mechanically removing vegetation removal (Environment Canterbury Regional Council, 2015).
- 240 Environmental flow management plans aim to mitigate someare aimed at mitigating undesired channel adjustments due to dam operations. Due to Given the costcosts of these programs, decision-makers are increasingly requesting the scientific

community to develop appropriate tools able tocapable of (i) identifyidentifying and controlcontrolling the factors that cause channel alterations, and (ii) to assessof assessing the effectiveness of management programs. Environmental agencies in several countries require dam operations to respect releasing protocol in an attemptrelease protocols to mitigate adverse
impacts on downstream ecosystems (Schmidt and Wilcock, 2008; Olden and Naiman, 2010; Watts et al., 2011; Konrad et al., 2012). Beisdes someAlthough successful empirical experiences do exist (Souchon et al., 2008; Konrad et al., 2011), robust predictive tools and models are becoming much more impellenturgently needed to predict channel responseresponses to dam operations and interruption of the longitudinal river continuum (Bliesner et al., 2009; McDonald et al., 2010; Melis, 2011; Coulthard and Van De Wiel, 2013; Gaeuman, 2014).

- 250 The assessment of future evolutionary trajectory of channel morphology maycan be achievedassessed using conceptual models (e.g., Channel Evolution Models – CEMs, as described in Schumm et al., 1984; Simon and Hupp, 1986; Simon, 1989), empirical (Lane, 1955; Schumm, 1977; Rhoads, 1992) orand numerical models, either Computational Fluid Dynamic (CFD) models or Reduced Complexity Models (RCM)-) (Larsen et al., 2014). Previous applications of RCMs on braided rivers have focused mainly on theoretical scale-independent analysis (Murray and Paola, 1994), laboratory experiments (Doeschl-Wilson
- and Ashmore, 2005; Doeschl et al., 2006; Nicholas, 2010), or short gravel-bed river reaches (Coulthard et al., 2002; Thomas and Nicholas, 2002; Coulthard et al., 2007; Thomas et al., 2007; Van De Wiel et al., 2007). In this study, such as in Ziliani et al. (2013) and Ziliani and Surian (2016), an attempt has been made towe apply an RCM model at mesospatial (i.e., 5-50 km) and mesotemporal (i.e., 10–100 years) scales. In particular, the CAESAR LISFLOOD model (Bates et al., 2010; Coulthard et al., 2013) has been hereinis applied to a 7–km–long braided reach of the Piave River (Italy), one of the most heavily and historically regulated river systems in Italy.
  - We applied the CAESAR LISFLOOD model (hereafter C-L) to assess the morphological effects related toof two different kinds of flow regime management strategies: the first is characterized by yearly controlled floods with peaks able to transportcapable of transporting sediments; while the second with more infrequent andconsists of less frequent higher magnitude floods (i.e., floods with 5-year recurrence interval equal to 5 yearsintervals) released only when notable channel
- 265 narrowing is observed in the evolutionary trajectory. Both strategies have been developed according to two main criteria: (i) to maximize the flow regime "re-naturalization" 5a meaning that the "Controlled Flood" (CF) duration has to be set in orderof the controlled flood (CF) is designed to increase its yearly likelihood to occur approaching theapproach natural scenario conditions as much as possible; and (ii) the "feasibility" of the strategy, which is verified by the fact that the cumulative volume released per year has to be loweris less than the maximum volume stocked-in the reservoirs existing upstream offrom the studystudied reach.
- This paper aims to addressaddresses two main issues: (i) to what extent the effectiveness of controlled floods can be effective for the geomorphic recovery of a-strongly regulated braided river?, and (ii) can the suitability and reliability of the reducedcomplexity CAESAR-LISFLOOD morphodynamic model CAESAR-LISFLOOD be considered a suitable and reliable tool to reproduce the morphological evolution of a-large gravel bed riverrivers at the given mesoscales?. In the first section of the paper, we provide a brief description of the studied river reach. The second section presents the available data, the two models
  - 10

used-(i.e., the morphodynamic model-CAESAR-LISFLOOD; model (Bates et al., 2010; Coulthard et al., 2013), and the hydrological model; (Botter et al., 2007), and the criteria adopted forto design the scenario strategy-design. The third section presents the results concerningin terms of (i) the historical river reach morphological river reach adjustments, (ii) the flow regime alterationalterations, (iii) the CAESAR-LISFLOOD calibration; and (iv) the simulations of three different "Controlled Floods" (CFs) releases controlled flood release scenarios. Finally, we critically discuss the results and examine the strengths and weaknesses of CAESAR-LISFLOOD and the effectiveness of the flow management strategies under investigation.

**2 General setting of the study area**

**2.1 The Piave River basin**

280

The Piave River is located in north-eastern ofnortheastern Italy, and it. It flows for about 220 km from the Alps to the Adriatic Sea (Fig. 1). The basin area is about 3,900 km2, and its average elevation is about 1,300 m a.s.l. (maximum elevation is 3,364 m a.s.l.). The climate is temperate-humid, with an average annual precipitation of aboutapproximately 1,350 mm. Significant annual variations in theAnnual rainfall amount have been measuredvaried substantially over the 20th century, but without any statistically relevantsignificant trends (Surian, 1999).

- AsLike most of theItalian Alpine Italian rivers (Surian and Rinaldi, 2003; Surian et al., 2009; Comiti, 2012), the Piave River
   has suffered heavy human impact, which has altered the basin and the river channel dynamics (Surian, 1999; Botter et al., 2010; Comiti et al., 2011; Comiti, 2012). Especially during the 20th century, theThe Piave basin has experienced a rapid increase ofin anthropogenic exploitation byin the 20th century, with the construction of a series of dams and reservoirs (nowadays therebetween the 1930s and 60s. There are now 13 major reservoirs, (Botter et al., 2010) built along the main stem and some tributaries from the 1930s to 1960s. At present, a(Botter et al., 2010). A complex regulation scheme existsis in place
- 295 (for details see Surian, 1999; Botter et al., 2010), designed) to maximize production of hydroelectric power production and the provision of provide irrigation water (Fig. 1). Flow regulation alters both the flow duration characteristics and the volume of annual runoff in the river. The reservoirs and diversions along the river and its tributaries also affect sediment transport and supply.
- The Piave basin had also experienced has historically experienced strong changes due to land use modifications. Especially afterSince the 1950s, the development of industry and tourism boosted the abandonment of traditional agricultural and cropping activitiesproduction on the mountain slopes, causing have been abandoned largely because of the development of industry and tourism, resulting in natural reforestation in the upper parts of the basin (Del Favero and Lasen, 1993). In addition to the reductions in sediment supply due to trapping by dams and reforestation, intense in-channel gravel mining has also contributed to alterreducing sediment fluxes since the 1960s. Furthermore, humanHuman pressure on the river channel dynamics has also resulted from construction of bank protection structures and torrent control works. As, as a result of these bank protection works, at presentwhich the river can still move laterally, although the available width for planform shifting is narrower than its natural braided belt.

**2.2. Study reach**

The studystudied reach is ~7 km-long (Fig. 1) and is located between Ponte nelle Alpi and Belluno (the drainage area at 310 Belluno is 1,826 km2). In this reach the The morphology of the reach is mainly braided and wandering. The average slope of the reach is 0.47%, and the median surface grain size ranges between 18 and 32 mm (Tomasi, 2009). The active channel width ranges between 43 and 452 m, beingand is 241 m on average, while the fluvial corridor width, defined by the presence of Holocene fluvial terraces, ranges between 106 and 1,110 m, beingand is 672 m on average. Previous studies (Surian, 1999; Surian, et al., 2009; Comiti et al., 2011; Picco et al., 2016) have outlined that; over the last 200 years, the studystudied reach

315 havehas undergone notable lateral adjustments (narrowing up to 66%), but notwith no significant changes ofin channel pattern. In terms of bed level changes, twopatterns. Two phases have been identified: a in terms of changes in bed levels, first a phase of moderate incision (1970-1990s) followed by a more recent phase (1990s-2003/2007) during which the river has exhibited equilibrium or slight aggradation (Surian, et al., 2009; Comiti et al., 2011).

**3. Materials and methods**

**320 **3.1 Channel morphology and reconstruction of** its evolutionary trajectory**

Channel morphology was analysed in order to gather (i) input data for the CAESAR – LISFLOOD model, (ii) data for model calibration, and (iii) evidence of the evolutionary trajectory of the studystudied reach. RiverThe river channel, islands, flowing ehannelchannels, bank protection structures and groynes were digitized using the available aerial photos and terrain models covering the studystudied reach (i.e., 2003, 2009 - Table 1). The analysis was carried out usingemployed ArcGIS 10.2. The flowing channels and the unvegetated or sparsely vegetated bars with little or no vegetation were merged to obtain measurement of measure channel width. BraidingThe braiding index was calculated using the average number of anabranches across the river (Ashmore, 1991; Egozi and Ashmore, 2008). The historical analysis earried out byof Comiti et al. (2011), which covered the period 1805 – 2006 has been, was extended up to 2009.

A LiDAR Digital Elevation Model (DEM) was provided by the Autorità di Bacino delle Alpi Orientali. It was created using
an airborne LiDAR survey that was aequired in July 2003 (orthometric elevations adopted, vertical error estimate ±20 cm) almost contemporary to one of thean aerial photosphoto used in this study (Table 1). Even though the river reach is characterized at low flow by the presence of rather small inundated areas, it was not possible to obtain bed elevation in the flowing channel areas with the standard LiDAR data. Therefore, to complete theTo obtain bed elevation extraction, the water depth was estimated through the application ofby applying the method proposed by Bertoldi et al. (2011) using the 2003 aerial
photos. This is an optical remote sensing technique (Marcus, 2012) for retrieving shallow water depth information using the color of the pixel, as, Legleiter et al. (2009) demonstrated that the log transformation of the green over red band ratio correlates linearly with water depth across a wide range of substrate types. The linear regression is usually should be calibrated by direct

measurementsmeasurement of water depths at the time of the aerial survey. Since such data were not available, we calibrated the regression coefficients by referring to the topography of boththe 2003 and 2009 cross-section surveys.

340 Sediment grain sizes were surveyed in 2009 (Tomasi, 2009) using volumetric sampling of the surface layer (Fig. 1). A single probability density curve was extracted (D50 ~ 24.5 mm, D15 ~ 2 mm, D84 ~ 77 mm) with nine size classes, as required by the C-L morphodynamic model.

**3.2 Analysis of the hydrologic regime inof the Piave River**

A variety of There are several approaches is available to analyse the impact of river regulation on the natural flow regime of
 rivers (Richter et al., 1996; Richter et al., 1997; Martínez Santa-María et al., 2008; Yin et al., 2015). In the case of the Piave
 River, although several studies have investigated the degree of alteration to its hydrological regime alteration (Villi and Bacchi, 2001; Botter et al., 2010; Comiti et al., 2011), such analysis washas been hampered by (i) the unavailabilitylack of a long\_term flow data series\_and (ii) the difficulty in sortingdistinguishing between natural and artificial componentcomponents of the flow regime. In Da Canal et al. (2007) and Comiti et al. (2011), flow records derived from two gauging stations (Busche

- and Segusino; Fig. 17) were modified using a specific corrective factor (Villi and Bacchi, 2001) and then merged. Comiti et al. (2011) confirmed that the largest flood event at Busche (Fig. 1) occurred in 1966 and reached almost 4,000 m3s-1.
   Furthermore, their analysis showed that the dischargedischarges with Recurrence Intervala recurrence interval (RI) of 2 years waswere not statistically different if calculated separately for pre- and post-regulation periods (1954 was used as the separation date between the periods). However, higher frequency the peak discharges of the more frequent events (RI ≤ 1.5 year) show a
- 355 reduction of peak dischargedecreased after 1954. Similar outcomes have also been reported in Picco et al. (2016). reported similar outcomes.

Botter et al. A(2010) offered a more detailed analysis of the impact of regulation on river regimes has been presented by with the application of Botter et al. (2010), who applied a physically-based modelling approach to assess the alterations of in the streamflow regime observed in various cross sections of the drainage network downstream offrom dams and weirs. The authors

- 360 have applied an analytical stochastic model (Botter et al., 2007) to characterize the streamflow probability density function (pdf) by means of climate, soil and vegetation parameters. After applying a preliminary model application to smaller, unregulated sub-catchments (that allowed tofor properly verifyverifying the capability of the model to reproduce locally the natural streamflow regime) locally, the authors haveauthor applied the model also into several regulated sections of the Piave River, including Soverzene (about 5 km upstream offrom Ponte nelle Alpi, Fig. 1), in order to evaluate the natural flow regime
- 365 in regulated cross-sections and, bybased on the difference, the effect of regulation on the statistical hydrographic features of the hydrograph. The approach conceptualizes the dynamics of daily streamflow as a sequence of peaks in response to rainfall and decays in between these jumps. These jump-decay dynamics are then linked to a catchment-scale soil-water balance wherein which the input is represented by stochastic daily rainfall. In this setting, flow-producing rainfall events (that lead to streamflow jumps) result from the censoring operated by catchment soils on daily rainfall, and theywhich are modeledmodelled
- 370 as a marked Poisson process with mean depth  $\alpha$  and mean frequency  $\lambda$ . The parameter  $\alpha$  identifies the average intensity of

daily rainfall events, while  $\lambda$  is the frequency of flow-producing events, which is smallerless than the underlying precipitation frequency because of the effect of soil moisture dynamics and evapotranspiration. As a consequence, several climate variables (such as rainfall attributes), as well as soil and vegetation properties-were, are embedded in  $\lambda$ . Additionally, in thatIn this framework, streamflow recessions in between flow pulses are assumed asto be exponential, with a mean rate equal to k, which defines the inverse of the time scale of the hydrological response (i.e., the mean water retention time in the upstream catchment). Under these assumptions, it can be shown that the steady-state pdf of the specific daily discharge (discharge per\_ unit catchment area) is a Gamma distribution with shape parameter  $\lambda/k$  and scale parameter  $\alpha k$ . The model is applied at the seasonal timescale, and then the annual pdf is calculated as the average of the four seasonal distributions. During winter, the The

- presence of snow dynamics in winter in the uppermost regions of the catchment is accounted for by reducing the size of the active contributing catchment and increasing the recession rates as described by Schaefli et al. (2013), with an elevation threshold of about 1,900 m a.s.l. In spring, a base flow value is added to the modeledmodelled streamflow distribution, which corresponds to a rigid rightward shift of the pdf. The probability distribution of the natural daily streamflow estimated by the model is then compared to the pdf of the observed daily flows to assess the extent of the impact of regulation in the lower reaches of the Piave River, and to get someobtain guidelines for devising meaningful strategies of the flow regime management
- 385 strategies. In particular, the daily streamflow series used in this study has beenwas recorded from 1995 to 2009 at Belluno gauging station located at the downstream section of the studystudied reach (Fig. 1). The highest flood event peaks observed in the reference periods (1996, 2000 and 2002) were checked and modified combining data atfrom Belluno and discharge measurements at the Soverzene weir (Braidot, 2003).

**3.3 The CAESAR-LISFLOOD model**

- Over the last 20 years the The application of hydro-morphodynamic physically-based numerical models (generally known as Computational Fluid Dynamic models, CFD) over the last 20 years has mainly been focused on laboratory idealized channel configurations at the laboratory scale (Wu et al., 2000; Defina, 2003; Rüther and Olsen, 2005; Abad et al., 2008) or referred to the morphological dynamicdynamics of natural channels over short time periods (Darby et al., 2002; Chen and Duan, 2008; Li et al., 2008; Wang et al., 2008; Zhou et al., 2009). Although their recent development, the restriction of their field The limited range of application of these models reflects unsolvedunresolved issues in terms of data availability and high computational demands (Escauriaza et al., 2017). Only a few recent works (i.e., Nicholas, 2013a; Williams et al., 2016) have shown that CFD models can be applied at larger spatial and temporal contextsscales. This limitation has drivenled to develop the development of alternative two-dimensional alternative models that have beenare commonly referred to as cellular automata (Murray, 2007), cellular models (Murray and Paola, 1994; Coulthard et al., 2002; Thomas and Nicholas, 2002; Coulthard et al., 2007; Parsons and Fonstad, 2007; Van De Wiel et al., 2007), exploratory models, andor reduced-complexity models (RCM Murray, 2007; Nicholas et al., 2006; 2012). These models have a common solution to the problem that, which is the adoption of simplified
  - 14

hydrodynamic and sediment transport equations derived by the abstractions of the governing physics.

|---------------------------------------|
|                                       |
|                                       |
|                                       |
|                                       |

[revised manuscript text omitted]
 Water management of historically regulated, water management rivers may be oriented to restorerestoring the flow regime to close to the prior that before impact conditions, typically aiming to reactivate aimed at reactivating physical processes linked to specific components of the flow regime (Wohl, 2011). Nevertheless, existing priorities in uses of the water resources use often limit the feasibility and the effectiveness of any flow regime re-naturalization strategies, and in most cases the strategy is merely reduced to the definition of a minimum volumesvolume of water released for partial restoration goals. Olden et al. (2014) provided a systematic review of systematically reviewed flood experiments to evaluate globally the success of this practice in flow regime management. They considered 113 flood experiments acrossin 20 countries

- were reviewed revealingand found that only 11% of the case studies were aimed directly toat morphological effects and about, while around 80% of the experiments involved only low magnitude flow events. Major part of Most experimental flow releases focusses on a biological variable variables (primarily fishesfish), aquatic organisms (Konrad et al., 2011) and reestablishing vegetation reestablishment (Shafroth et al., 2010), rather than on abiotic factor factors (e.g., channel morphology)
   (Wohl et al., 2015b). The so-called What are termed "environmental flows" and "Green Hydro" are concepts widely accepted
- even though theyconcepts that refer mainly to quantity, timing, duration, frequency and quality of water flowsflow releases,

as required to sustain freshwater, estuarine and near-shore ecosystems, according to social interests (Acreman and Ferguson, 2010; Rivaes et al., 2017).

- Thus, there arehave been few examples of flow regime recovery strategies which have been designed on a geomorphological basis or rather planned to with primarily achieve-morphological targetsgoals. The Colorado River below the Glen Canyon Dams represents the most relevantimportant exception: several. Several controlled floods (i.e., five High Flood Experimentshigh flood experiments) were released in thebetween 1996-and 2013-period to maintain and rehabilitate sandbars that occur in lateral flow separation eddies (Schmidt and Wilcock, 2008; Melis, 2011; Mueller et al., 2014). Also the The Trinity River (California, USA) represents ananother excellent case study, as a in which morphological goals were among the multiple objective flow designed included morphological issueobjectives (Trinity Management Council, 2014). Still, Nevertheless, with the exception of the lower Waitaki River in New Zealand, there are no experiences are available on examples of morphological recovery or conservation of braided rivers, with the exception of the Lower Waitaki River (New Zealand), even thoughand no controlled floods have been used in that riveron the Waitaki (Hicks et al., 2003; Hicks et al., 2006; Environment Canterbury
- Regional Council, 2015).
   515 Controlled Provoking controlled floods andor removing vegetation removal actions are expensive costly and potentially foregocan preclude other positive feedback effects. It is therefore worth to preliminary evaluate and testactions, so it worthwhile to assess these kinds of actions taking into accountstrategies and consider the morphological effects of different channel-forming discharges (Surian et al., 2009). As emphasized by Rathburn et al. (2009); stressed that flow releases from dams must exceed a series of thresholds to be morphologically effective. Four discharge thresholds of increasing threshold
- 520 dischargesintensity should be identified in a morphological recovery/conservation plan, each able to activatecapable of activating a specific morphological process: (i) the mobilization of interstitial sediment essential for the hyporheic exchange maintenance; (ii) the mobilization of the streambed to maintain natural bedforms; (iii) the inundation of overbank units (i.e., berms, floodplains, terraces) to confine encroachment by xeric plants; and (iv) the lateral channel mobility that may promote the removal ofcan remove senescent woody vegetation and create opportunities for seedlings to germinate and mature. Once each threshold discharge value or range is quantified, the flow duration can be tuned within the limits imposed by the

actualavailable flow availability, assuming, using the natural flow regime as a reference.

In this work, the seenarios of The flow regime management scenarios used in this work were defined referringin reference to three flood threshold levels that partially matchingmatch those proposed inby Rathburn et al. (2009): (i) thefull in-channel full transport discharge assumed to be able to mobilize thecapable of mobilizing interstitial sediment, (ii) the bankfull discharge

- 530 able to maintain thecapable of maintaining natural bedforms; and (iii) the overbank discharge able to affect apable of affecting the main lateral units by inundation. A field study conducted in the reach using painted sediments similar to (those used in Mao and Surian., (2010;) and Mao et al., (2017) allowed toresulted in an estimate that of the first reference threshold is about at 80 m3s-1 (RI < 1 year, full transport discharge able to mobilize apable of mobilizing sediments, irrespective of their size). The bankfull discharge was estimated using the calibrated C-L model and was approximated to the discharge filling the
- 535 active channels and bars without overflowing onto the oldest island assumed to be morphologically equivalent to the recent

fluvial terraces (Williams, 1978; Pickup and Rieger, 1979; Surian et al., 2009). The bankfull discharge is about 500 m3s-1, and corresponds to a 2.5 years RI, coherentlywhich is consistent with other literature references in the literature (Leopold, 1994). Only the > 1,000 m3s-1 floods were identified to be able to a capable of completely inundate inundating the oldest island and overflowing locally overflow into the recent fluvial terraces.

- 540 Four simulation scenarios, each were considered, all covering the same 25 years long-year period (2009-2034), were explored) (Table 2). The first scenario (i.e., "baseline scenario", SC1) eorresponds tois the current condition, characterized by a strongly altered flow regime. In this case, the discharge series measured infor 1995-2009 (hourly data at Belluno gauge station) has beenwas repeated twice. ScenariosBoth scenarios 2 (SC2) and 3 (SC3) were both set up usingused a flow regime strategy characterized by one CF per year. In SC2 theThe yearly CF in SC2 had a constant value of 135 m3s-1 (RI ~ 1.08 years). This
- 545 value has been), which was calculated as the average of the maximum annual floods observed in 1995-2009,. This was higher than the reference threshold discharge, i.e. the full in-channel <del>full</del> transport discharge (80 m3s-1). In SC3, the The</del> yearly CF values for SC3 were randomly selected above the threshold discharge using the natural streamflow pdf estimated by the model (minminimum value = 80 m3s-1; maxmaximum value = 276 m3s-1, RI ~ 1.4 years). In this case, the average <del>value</del> of all the CF values was found to beis equal to the SC2-yearly CF discharge for SC2, assuming values included in the [80 276] m3s-1 range.
- 550 All the CFs had a fixed duration of 5 days, according to the re-naturalization maximization criteria. In SC2 and SC3, the cumulative likelihood to occur associated towith the released peaks raised from 0.025 (actual observed altered regime) to 0.04, being the natural reference value being 0.14. Scenario 4 (SC4) was planned to represent a different management strategy, consisting inof larger CFCFs released by dams (constant value for one-day) only following the observation of notable channel narrowing. Specifically, we assumed 200 m as a threshold for average channel width,
- 555 considering the evolutionary trajectory over the last 200 years and, in particular, the most intense narrowing that took place in the early 1990s (Fig. 3). Taking into account theBased on channel width measurements eonductedtaken by a one-year step duringin the SC4 simulation, only two CFs were released, one in 2020 and the other in 2032-respectively. The released discharge was fixed equal toat 600 m3s-1 (RI 5 years), ranging between the second and the third reference threshold discharges, discussed above, so that it was surely able to maintain the would undoubtedly be capable of maintaining in-channel.
- 560 bedformsbedform dynamics, while completely avoiding any hydraulic risk issues risks and damages in the overbank units (i.e., j recent terraces), locally occupied by secondary roads and cultivated fields. All the CFs have been were released in November avoiding to overlap existing, while overlapping floods, were avoided. All J

the scenarios made use of the same model setting-achieved, which was obtained in the calibration phase, using the 2009 calibration run output data in raster format as the initial boundary conditions (i.e., bed elevation, not erodibleunerodible, by hydraulic structures, vegetation cover, grain size bed sediment distribution). As reported in Table 2, all the scenarios required

the release of a cumulative volume to be released per year (i.e., per flood)), which was considerably lowerless than the *f* maximum stocked volume in the reservoirs existing upstream offrom the studystudied reach (90.8 Mm3). SC2 and SC3 required *f* similar volumes of about 58 Mm3 of water, corresponding to about or around 1.45x 103 Mm3 onover the whole entire scenarios

|---------------------------------------|

period. SC4 represents the cheapestlast expensive scenario indeed because it needed 51.8 Mm3 per year and 104 Mm3 globally, one order of magnitude lowerless than in the other scenarios.

**4. Results**

570

**4.1 Evolutionary trajectory of channel morphology**

Using the available dataset on morphological changes of the Piave River (Comiti et al., 2011), we reconstructed and updated the channel adjustments up to 2009. The analysis focused on two sub-reaches, respectively upstream and downstream offrom 575 San Pietro in Campo, where the river is naturally more confined (Fig. 1). The division in reach was divided into two subreaches was used to better describe the morphological adjustments over the 1800-2009 period. The trends in average width trends are similar for bothin the sub-reaches are similar, and are characterized by four main adjustment phases (Fig. 3): (i) a first period (during the 19th century and the first half of the 20th century) dominated by a braided pattern, channel width equal to about the 80% of the alluvial plain width, negligible morphological changes and the absence of a dominant process (i.e., channel widening or narrowing), (ii) a second phase of adjustment with channel narrowing of about 60% from the 1950s to 580 the early 1990s (whole reachthe channel width of the entire reach was 370 and 247, respectively in 1960 and 247 in 1991), interrupted by a large flood event in 1966 (RI ~ 200 years - (Comiti et al., 2011), which caused a temporary widespread channel expansion; (iii) a phase of channel widening during the 1990s (whole reachthe channel width of the entire reach was 342 m in 1999)), mainly related to the 1993 flood event, characterized by a 12-years-year RI (Comiti et al., 2011); and (iv) the 585 most recent adjustment phase characterized by channel narrowing. Focusing on the last 20-25 years, after following the 2002 flood, the river turned toentered a new phase (IV in Fig. 3) characterized by narrowing: channel width in 2009 (i.e. 241 m) was the lowest value observed in the studystudied reach over the last 200 years. While widening during phase III-widening was likely due to the termination of in-channel gravel mining (Comiti et al., 2011), the most recent phase of narrowing (i.e.,

**590 4.2 Flow regime alterations**

TheA comparison betweenof the frequency distribution of observed daily streamflows at the\_Belluno cross—section and the model-based estimate of the streamflow pdf under unregulated conditions (Fig. 4) shows the extent of the significant impact of regulation in the lower reaches of the Piave River. The mean and the mode of the streamflow distribution are significantly reduced by the anthropogenic exploitation of water resources (i.e., by-pass flows and diversions). Accordingly, the exceedance probability of moderate to high flows is significantly reduced under current regulated conditions. In particular, the probability to observe discharges larger than 80 m3s-1 is reduced by about one order of magnitude (i.e., from 0.14 to 0.025). Such results are crucial for setting the flow-regime management scenario since (i) they show that a strategy aming to improve and at improving the current flow-regime should be implemented is needed. (ii) this strategy should compensate the expected low

phase (phase IV) was likely due to the absence of major floods (see also Fig. 2).

morphological dynamism of the river caused by the decreased occurrence of discharges able to mobilizecapable of mobilizing sediments and produceproducing significant morphological changes in the studystudied reach.

It is worth to notenoting that the hydrological model underestimates the frequency of the highest flows (i.e., discharges largerof more than 300 m3s-1) because all the non-linearities of the hydrologiehydrological response (e.g., the presence of different flow components such as surface runoff) are neglected in this version of the model (Basso et al., 2015). As a consequence, the probability associated towith the highest flows in regulated conditions is largergreater than the corresponding value estimated by the stochastic model for the natural setting. This model However, this limitation, however, of the model does not bearhave any significant consequence for the analysis carried out in this paper, provided that the frequency of such high flows is relatively low.

**4.3 Calibration of the morphodynamic model**

The results presented inof Ziliani et al. (2013) and Coulthard et al. (2013) have been takenwere used as a reference to 610 achievecalibrate the C-L ealibration model. According to the results of the sensitivity analysis in Ziliani et al. (2013), the lateral erosion rate and maximum erosion limit have been assumed aswere the most sensitive factors that required accurate tuning. The other factors (see Table 3), including the main new parameters introduced in the C-L version, were tuned manually through a "by trial- and- error" ealibration strategy (n. (75 runs in total). Following the performance evaluation techniques used by Ziliani et al. (2013), the calibration was based on performance indices developed specifically for data available in a raster 615 format (Bates and De Roo, 2000; Horritt and Bates, 2001). The performance indices reported in Table 4 were calculated for all the calibration runs at the end of the simulation (2009), that is these being (i) the vegetation performance index ( $F_{veg}$ ), (ii) the wet area performance index (Fwet) and (iii) the active channel performance index (Fc). In addition, several planimetric features have been were calculated, including (i) average active channel width, (ii) equivalent wet area width (Lw), and (iii) the mean braiding index (Egozi and Ashmore, 2008). The results (see Table 4, Fig. 5, Fig. S3 and S4 in the "Supplementary 620 material" fileMaterial File) show a "very good performance" (performance class as defined in Henriksen et al., 2003; Allen et al., 2007) for both the vegetation cover ( $F_{veg}$  69.7%) and the active channel area ( $F_c$  54.2%). Output values of the active channel width and braiding index values confirmed these results. The difference between the real and modeled 2009 active channel width (6 m) is lowerless than the input DEM cell size (10 m), and the modeled modelled braiding index value (1.71) is very close to the real value (1.69). The model performance is poor only performed poorly in reproducing the flowing channel 625 position (Fw 15.7 %)%), which partially confirming confirms the results presented inof Ziliani et al. (2013).

In order to integrate the morphological performance evaluations, we carried out an estimation of the We estimated mean annual bed load sediment yield at the downstream end of the reach and along the wholeentire reach. In the 2003-2009 period, the modeled average to integrate morphological performance evaluations. Average modelled bed load sediment yield resulted of aboutin the 2003-2009 period was around 21.5 x 103 m3yr-1. ModeledModelled yield varies significantly along the reach (up to 30%) taking%), with higher yearly values in the sub-reach upstream San Pietro in Campo. Significant differences exist between the maximumMaximum and minimum annual values. differ significantly. The 2006 minimum eorresponds to an

average annual sediment yield ofis about 260 m3yr-1 versus the 2008 maximum of about 53.3 x 103 m3yr-1. Such These sediment transport values agree with estimates for gravel-bed rivers with similar characteristics to those of the Piave River reach (Martin and Church, 1995; Ham and Church, 2000; Nicholas, 2000; Liebault et al., 2008; Ziliani et al., 2013; Mao et al., 2017).

**635 4.4 Channel response to flow regime management strategies: scenario results**

ChannelThe channel adjustments induced by allthe different scenarios were assessed by comparing every year (in February) the active channel width and the braiding intensity (BI) in each year (in February) using the same techniques adopted in the calibration phase (Fig. 6). Channel width in Scenarios 2-4 was almost always highergreater than in SC1 (the "baseline scenario"). On average, during the whole scenario period, SC2 and SC4 produced comparable). Average channel widening of

- 640 aboutin SC2 and SC4 were similar throughout the scenarios, being approximately 6.4% (~ 14 m), while at the end of the scenarios (2034), widening was about 9% (~ 25 m) and 13.5% (~ 38 m) in SC2 and SC4, respectively. The SC3 induced **a** slightly lowerless widening, about 5.4%, duringthroughout the whole period, and 8.6% (~ 24 m) at the end of the simulated period. The maximum annual widening was observed in SC4 (~ 120 m in 2033), followed by SC3 (~ 77 m in 2020) and SC2 (~ 43 m in 2032 Fig. 7).
- 645 Results suggest that the CFs scenarios (SC2-4) and the baseline scenario (SC1) provide similar long-term morphological trajectories characterized by alternate phases of widening and narrowing and notable changes in active channel width (width varies between 150 and 360 m). Figure 7 shows that each channel width oscillation takesoscillations take place in about-over a period of 6- to 7 years and it has 160 man amplitude of 160 m in response to the alternation of periods characterized by different magnitude floods series: in the 2011-2015 and 2022-2028 periods, during which seven floods > 400 m3s-1 (RI ~ 1.9 years) occur, channel width follows a quasi-steady trend and is largergreater than 300 m. Instead, channel width decreased during the following periods (2017-2021 and 2029-2031) affected by lower magnitude floods (200 m3s-1 maximum peak value), channel width shows decreasing trajectories. Over the whole 25 years), SC1 provides had a slightly decreasing trend
- (Fig. 7) that is not reversedover the entire 25 years (Figure 7). Channel width has quasi-zero slopes in the otherCF scenarios.
   In all the CFs scenarios the channel width trend assumes quasi-zero slope, even if the channel width measured at the end of
   the simulations invalues is about [8.6-13.5 %] highermore than width in the baseline scenario.
- The braiding index inindices of Scenarios 2-4 waswere similar; to or lower; than inthat of SC1. SC4 resultedwas the most similar scenario elosest to thethat of SC1, with aan average BI in-time averaged value equal to of 2.78, onlywhich was slightly lower than thethat of SC1 (-1.5%). During SC2 we measured relativeDifferences in BI were higher differences in SC2 than in the BI value compared to SC1 (-7.3%, 0.21 BI unit Fig. 7), although these differences are quitewere small. In termsThe behaviour of trajectory, braiding intensity shows awas different behaviour in comparison to from channel width, as in that there are no clear oscillation phases butwas one period (from 2009 to 2023) with ano clear\_oscillations, and a clearly increasing
- trend, followed by a decreasing or quasi-steady (SC1) period until the end of the simulation. There is a non-linear correlation between BI and flooding series magnitude or the CFs. In particular, SC1 is the only scenario that doesdid not show an inverse trend inversion after 2023, and while the SC2 scenario has a very anomalous trend showing a anomalously had a consistently

665 lower BI value steadily lower that the other CFsCF scenarios, while SC3 and SC4 show a good agreement in theirhad very similar BI trendsyalues.

**5. Discussion**

**5.1 Geomorphic effectiveness of controlled floods**

Comparing the Several insights can be obtained from comparing future scenarios to the historical evolutionary trajectory (Fig. 670 3) several insights can be obtained3), despite the evident mismatch between the temporal frequency of the past and future channel width series (one value every 16.5 years in the 1805-1970 period and every 6.5 years in the 1970-2009 period; yearly values for the future series). It can be observed that: (i) the maximum channel widths reached duringin all-of the four future scenarios (in the periods 2015-2016 and 2028-2029) are close to the width in 1999, (ii) the minimum widths achieved in all future scenarios (2020 and 2032);) are always belowless than 185 m (with the exception of the first minimum duringin scenario 3) and are significantly lower than the historical minimum observed in 2009 (241 m), (iii) albeit with a low confidence level, we can state that the trajectory between 1991 and 2009 (phase III and IV described in Par.Section 4.1) seems to followfollows an oscillatory oscillating evolution with half the frequency of the oscillation modeled modelled between 2009 and 2034, (iv) thethere is a good correlation between the variation of channel width and the flow regime reproduced forin the future scenarios, whereas this is not always straightforwardthe case in the past evolution. This point is exemplified by the rather major 2002 680 flood event, which did not re-widen the river at the levels of 1999 (about 342 m), despite being relevantsignificant in terms of magnitude (13-years-year RI). Indeed, the width the following year (2003) the width was approximately 289 m, about 15% less than in 1999. This may suggests uggests that the study reach, after a period (phases I and II in Fig.Figure 3) of morphological instability characterized by a prevalentnarrowing tendency to narrowing, has reached, the studied reach had acquired a new morphological equilibrium configuration characterized by a periodic oscillations ofin channel width. Similar 685 new equilibrium conditions, mainly controlled by the flow regime (i.e. frequency and magnitude of formative discharges) and

vegetation establishment, have been observed in the Tagliamento River (Ziliani and Surian, 2016). The intercomparison comparison of our four simulations shows that a few high magnitude floods provide slightly better morphological recovery/conservation than small yearly floods, <del>alsoand</del> at a significantly lower operational cost. Therefore, SC4 should be preferredis preferable to SC2 and SC3 from a <del>purepurely</del> morphodynamic point of view. Nevertheless, <del>results</del>

- 690 suggestthe comparison suggests that (i) none of the CFsCF scenarios are able to change can significantly thechange long-term channel width and braiding intensity trends, (ii) CFs releaseCF releases have no significant morphological benefits and do not represent a solution for athe morphological recovery inof braided rivers that suffered such strong and historical impacts in terms of flow and sediment supply regimes. It is worth noting that the selected CFs are feasible, that is taking into account the water infrastructure in the Piave River basin, and it is unlikely that higher or more frequent floods could be released. These results partially confirm the outcomes of Hicks et al. (2003) referring toon the Waitaki River, a gravel-bed river with similar
- characteristics to those of the Piave River. The authors state that, if a wider and more active channel is desired, an approach

eonsisting in the frequent release of "channel maintenance floods" from dams should be pursued-if wider and more active channels are desired. Hicks et al. (2013) showshowed that this kind of strategy may behas been unsuccessful and only multiyearsycar high magnitude CFs can produce temporary stable effective channel widening channel condition.

- 700 The cost of CFs is probably smallerless than those that of alternative strategies focused on increasing sediment supply, such as sediment augmentation, because flood releases commonly can be performed without redesign of often do not require redesigning reservoir structures. Nevertheless, reintroduction of flood flows implies "loss" of resourceresources stocked for other purposes (e.g., hydroelectric production, and supply of drinking or irrigation water-supply). Another feasible way for sediment augmentation is the removal (at least in part) of non-strategic bank protections along the reach. However, as suggested by Picco et al. (2016);) suggested, this kind of strategy should be preventively assessed last resort since these structures are
- still viewed by local populations as necessary to protect riparian woodlands that are highly appreciated for recreation and timber production.

Overall, this work gives useful insights for the Piave River management and, in general, for management of braided rivers with heavily impacted in-flow and sediment regimes: (i) none of the tested controlled flood strategies that was tested is able
 tocan significantly change-the on-going morphological evolution; (ii) the baseline scenario, without controlled flood releases (i.e., the no action strategy), provides a similar morphological evolutionevolutionary trajectory similar to that induced by the controlled floodsflood release scenarios. Therefore, a main outcome is that controlled floods (including high-magnitude floods, e.g. 5-year RI) may not have any significant effects on regulated rivers, specifically if formative discharges have been strongly altered.

**715**

**5.2 Assessment of CAESAR-LISFLOOD performance**

In-Ziliani et al. (2013) the authors-concluded that the main factors causingreasons for the poor morphological poor response of CAESAR are (i) the-DEM cell size, aswhich has been pointed out-also in others works (Doeschl-Wilson and Ashmore, 2005; Doeschl et al., 2006; Nicholas and Quine, 2007), (ii) the quality of data (i.e., lack of wet channel topography), and (iii)
the low flow periodsperiod removal, and therefore the eut-offelimination of the consequent morphological "gardening" phenomena (Ziliani et al., 2013). The combination of these factors produced a smoother and simpler braided morphology. The Piave case study represents an effort to achieve a better performance by (i) the flow routinesroutine refinement included in the LISFLOOD-FP module (one of the most recent and advanced Reduced Complexity Hydraulic Model schemereduced complexity hydraulic model schemes), (ii) the adoption of input data of higher quality input data (higher resolution DEM, bathymetry and hourly boundary conditions) and (iii) the code conversion in parallel programming methods. The results lead to an overall improvement of the model performance considering (i) the good channel width performance in the calibration phase, (ii) the excellent reproduction of braiding complexity-reproduction, including the pioneerpioneering and complex islandsjsland dynamics, both in the calibration and inthe long-term simulations, (iii) the reasonable estimation of bedload transport, and the small changes in bed grain size in the long-term simulations (e.g. D50 changed from 24.9 mm to 22.7, 24.3,

24

730 23.6 and 23.1 mm respectively in SC 1, 2, 3, and 4), and (iv) the adequate computation speed, close to the expectations (i.e., what was expected (about 10 days of computation for 25 years of hourly series).

The suitability of the RCMs application for the investigation of investigating river dynamics has been discussed in several previous studies (Doeschl-Wilson and Ashmore, 2005; Brasington and Richards, 2007; Nicholas and Quine, 2007; Murray, 2007; Nicholas, 2012, 2013b; Ziliani et al 2013; Ziliani and Surian, 2016). A general conclusion of these works is that RCMs

- 735 may provide morphological responses both unrealistic and highly sensitive morphological responses to model grid resolution. These problems are commonly interpreted as a direct consequence of both the adoption of flow routing schemes that neglect the momentum conservation and the use of local bed slopes for thein calculating bedload transport calculation (e.g., through the application of the uniform flow approximation).
- The C-L model maycan be considered ana useful tool in the search of an effective combination of simplicity and physical realism in the context of reduced-complexity modelling, overcoming some of the previous problems associated with earlier simplified hydrodynamic simplification issues.models. The encouraging results achieved inof this case study seem to justifysuggest the effort faced in such further development ofto develop this RCM. Although is justified. While the physical realism of flow and morphodynamic rules can remain unsolved at smallersmall scales (i.e., scales lower thanbelow the DEM cell dimension), the improvement superiority of the C-L model response at reach scales compared to over the older
- 745 CAESARearlier CESAR model is evident. Although the reduced at reach scale. Reduced-complexity modelling approach model will probably will not provide insights into some of the answers to key reductionist key questions eurrently faced byof hydraulic engineers and fluvial geomorphologists, nevertheless the model can provide useful insights for management. Specifically, insights, specifically about their macro morphological macromorphological river features (e.g. average channel width, braiding intensity) and adjustments (e.g. prediction of future evolutionary trajectory) of braided rivers.
- 750 The inherent limitation in limitations of reduced-complexity modelling approach doesmodels do not preclude the adoption of using RCMs where the aim is to represent meso-scale system behaviour at the mesoscale, rather than to makemaking reductionist predictions that are theoretically more accurate in quantitative terms. Reproducing the morphodynamic processes at each scale requiredrequires necessarily some forms of simplification, regardless of the level of complexity of the model adopted, and in CFD models as well. Nevertheless, this cannot be considered a convincing is not sufficient reason to necessarily
- 755 callingcall into question whether explanations the usefulness of river behaviour based on applying this kind of model have any application in the models in real world situation. In the light of the results presented inof this work and of the limitations faced anyway by theof reductionist alternative approaches (Williams et al., 2016), we believe that RCMs; and C-L model specifically, remain an the C-L model are attractive optionoptions for simulating river evolution over historical time periodshistorically and into the future scenarios, that is at the scale of interest for river management.
- 760 This work presents another case study in which an RCM has given realistic outputs in a large gravel-bed river, especially in terms of evolutionary trajectories. The suitability in reproducing macro morphological features and meso-scale processes should not be questioned any longer (Nicholas, 2013b). The capability to model small-scale phenomena remains open for RCMs as for all CFDs that try to reproduce phenomena deeply influenced by initial and boundary conditions, for which a data

[revised manuscript text omitted]

Basso, S.,